# Understanding and Mitigating Copying in Diffusion Models

**Gowthami Somepalli** [1], **Vasu Singla** [1], **Micah Goldblum** [2], **Jonas Geiping** [1], **Tom Goldstein** [1]

[1] University of Maryland, College Park
{gowthami, vsingla, jgeiping, tomg}@cs.umd.edu

[2] New York University
goldblum@nyu.edu

## Abstract

Images generated by diffusion models like Stable Diffusion are increasingly widespread. Recent works and even lawsuits have shown that these models are prone to replicating their training data, unbeknownst to the user. In this paper, we first analyze this memorization problem in text-to-image diffusion models. While it is widely believed that duplicated images in the training set are responsible for content replication at inference time, we observe that the text conditioning of the model plays a similarly important role. In fact, we see in our experiments that data replication often does not happen for unconditional models, while it is common in the text-conditional case. Motivated by our findings, we then propose several techniques for reducing data replication at both training and inference time by randomizing and augmenting image captions in the training set. Code is available at https://github.com/somepago/DCR.

## 1 Introduction

A major hazard of diffusion models is their ability to produce images that replicate their training data, often without warning to the user [Somepalli et al., 2022, Carlini et al., 2023]. Despite their risk of breaching privacy, data ownership, and copyright laws, diffusion models have been deployed at the commercial scale by subscription-based companies like *midjourney*, and more recently as offerings within search engines like *bing* and *bard*. Currently, a number of ongoing lawsuits [Saveri and Butterick, 2023] are attempting to determine in what sense companies providing image generation systems can be held liable for replications of existing images.

In this work, we take a deep dive into understanding the causes of memorization for modern text-to-image diffusion models. Prior work has largely focused on the role of duplicate images in the training set. While this certainly plays a role, we find that image duplication alone cannot explain much of the replication behavior we see at test time. Our experiments reveal that text conditioning plays a major role in data replication, and in fact, test-time replication can be greatly mitigated by diversifying captions on images, even if the images themselves remain highly duplicated in the training set. Armed with these observations, we propose a number of strategies for mitigating replication by randomizing text conditioning during either train time or test time. Our observations serve as an in-depth guide for both users and builders of diffusion models concerned with copying behavior.

## 2 Related work

**Memorization in generative models.** Most insights on the memorization capabilities of generative models are so far empirical, as in studies by Webster et al. [2021] for GANs and a number of studies for generative language models [Carlini et al., 2022, Jagielski et al., 2022, Lee et al., 2022].

37th Conference on Neural Information Processing Systems (NeurIPS 2023).

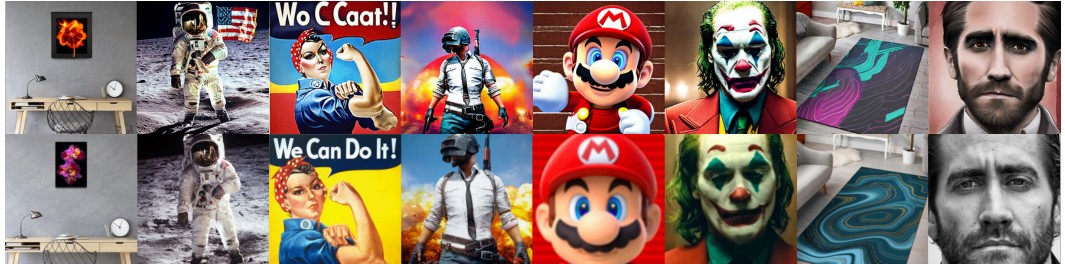

**Figure 1:** The first row shows images generated from *real user prompts* for Stable Diffusion v2.1. The second row shows images found in the LAION dataset. Please refer to Appendix for the corresponding captions.

Recently, Somepalli et al. [2022] investigated data replication behaviors in modern diffusion models, finding 0.5-2% of generated images to be partial object-level duplicates of training data, findings also mirrored in Carlini et al. [2023]. Yet, what mechanisms lead to memorization in diffusion models, and how they could be inhibited, remains so far uncertain aside from recent theoretical frameworks rigorously studying copyright issues for image duplication in Vyas et al. [2023].

**Removing concepts from diffusion models.** Mitigations deployed so far in diffusion models have focused on input filtering. For example, Stable Diffusion includes detectors that are trained to detect inappropriate generations. These detectors can also be re-purposed to prevent the generation of known copyrighted data, such as done recently in *midjourney*, which has banned its users from generating photography by artist Steve McCurry, due to copyright concerns [Chess, 2022]. However, such simple filters can be easily circumvented [Rando et al., 2022, Wen et al., 2023], and these band-aid solutions do not mitigate copying behavior at large. A more promising approach deletes entire concepts from the model as in Schramowski et al. [2023] and Kumari et al. [2023], yet such approaches require a list of all concepts to be erased, and are impractical for protecting datasets with billions of diverse images covering many concepts.

## 3 How big of a problem is data replication?

Somepalli et al. [2022] and Carlini et al. [2023] have shown that diffusion models *can* reproduce images from their training data, sometimes to near-perfect accuracy. However, these studies induce replication behavior by prompting the model with image captions that are directly sampled from the LAION dataset – a practice that amplifies the rate of replication for scientific purposes. We study the rate at which replication happens with *real-world* user-submitted prompts. This gives us a sense of the extent to which replication might be a concern for typical end-users.

We randomly sample 100K user-generated captions from the DiffusionDB [Wang et al., 2022] dataset and generate images using Stable Diffusion 2.1 [Rombach et al., 2021, Stability AI, 2022]. We use SSCD features [Pizzi et al., 2022] to search for the closest matches against these images in a subset of the training dataset. Note that SSCD was found to be one of the best metrics for detecting replication in Somepalli et al. [2022], surpassing the performance of CLIP [Radford et al., 2021]. We compare each generated image against $\sim 400$ million images, roughly $40\%$ of the entire dataset.

We find that $\sim 1200$ images ($1.2\%$) have a similarity score above 0.5, indicating these may be duplicates. Note that this is likely an underestimation of the true rate, as our SSCD-based search method is unlikely to find all possible matches. We show several of these duplicates in Figure 1. Sometimes, the captions precisely describe the image content (for example, 'Jake Gyllenhaal'). However, we also discover instances where captions do not mention the content from the generated image. For example, in the first duplicate, the user caption is "flower made of fire, black background, art by artgerm", and the LAION caption is "Colorful Cattleya Orchids (2020) by JessicaJenney". All the user and LAION captions for Figure 1 are in Appendix B.2. Several duplicates identified by SSCD also have high similarity scores due to the simple texture of the generated image. We include these SSCD false positives and other examples in the Appendix.

## 4 Experimental Setup

A thorough study of replication behavior requires training many diffusion models. To keep costs tractable, we focus on experiments in which large pre-trained models are finetuned on smaller

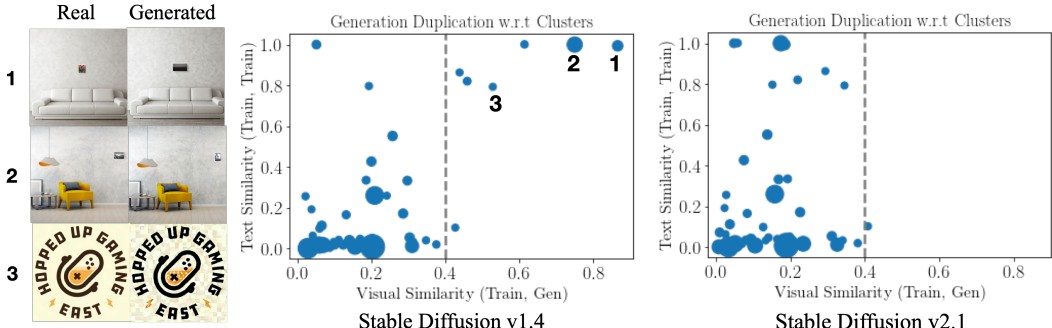

**Figure 2:** Stable Diffusion v1.4 generates memorized images when either images or captions are duplicated. Highly replicated generations from Stable Diffusion v1.4 and duplicated images from LAION training data are labeled on the plot and shown on the left. Stable Diffusion v2.1 is trained on a de-duplicated dataset, so as expected we see the clusters with high visual image similarity vanish from the right side of the chart. Nonetheless, we still see a number of replicated generations from clusters with high caption similarity.

datasets. This process reflects the training of Stable Diffusion models, which are pre-trained on LAION and then finetuned in several stages on much smaller and more curated datasets, like the LAION Aesthetics split.

**Datasets:** We use Imagenette[1], which consists of 10 classes from Imagenet [Deng et al., 2009] as well as two randomly sampled subsets of $10,000$ and $100,000$ images from LAION-2B [Schuhmann et al., 2022] for our experiments. The LAION subsets, which we denote as LAION-10k and LAION-100k, include captions, while the Imagenette data is uncaptioned. For some experiments, we use BLIP v1 [Li et al., 2022] to generate captions for images when needed. We provide details for LAION subset construction in the Appendix.

**Architecture & Training:** We use the StableDiffusion-v2.1 checkpoint as a starting point for all experiments. Unless otherwise noted, only the U-Net [Ronneberger et al., 2015] part of the pipeline is finetuned (the text and auto-encoder/decoder components are frozen) as in the original training run, and we finetune for 100000 iterations with a constant learning rate of $5e-6$ and 10000 steps of warmup. All models are trained with batch size 16 and image resolution 256. We conduct evaluations on 4000 images generated using the same conditioning used during model training. Refer to Appendix for complete details on the training process.

**Metrics:** We use the following metrics to study the memorization and generation quality of finetuned models. **Frechet Inception Distance (FID)** [Heusel et al., 2017] evaluates the quality and diversity of model outputs. FID measures the similarity between the distribution of generated images and the distribution of the training set using features extracted by an Inception-v3 network. A *lower* FID score indicates better image quality and diversity.

Somepalli et al. [2022] quantify copying in diffusion models using a similarity score derived from the dot product of SSCD representations [Pizzi et al., 2022] of a generated image and its top-1 match in the training data. They observe that generations with similarity scores greater than $0.5$ exhibit strong visual similarities with their top-1 image and are likely to be partial object-level copies of training data.

Given a set of generated images, we define its **dataset similarity score** as the 95-percentile of its image-level similarity score distribution. Note that measuring average similarity scores over the whole set of generated images is uninformative, as we are only interested in the prevalence of replicated images, which is diluted by non-replicated samples. For this reason, we focus only on the similarity of the 5% of images with the highest scores. Please refer to Appendix for a longer discussion on this metric design.

## 5  Data Duplication is Not the Whole Story

Existing research hypothesizes that replication at inference time is mainly caused by duplicated data in the training set [Somepalli et al., 2022, Carlini et al., 2023, Webster et al., 2023]. Meanwhile,

---

[1]https://github.com/fastai/imagenette

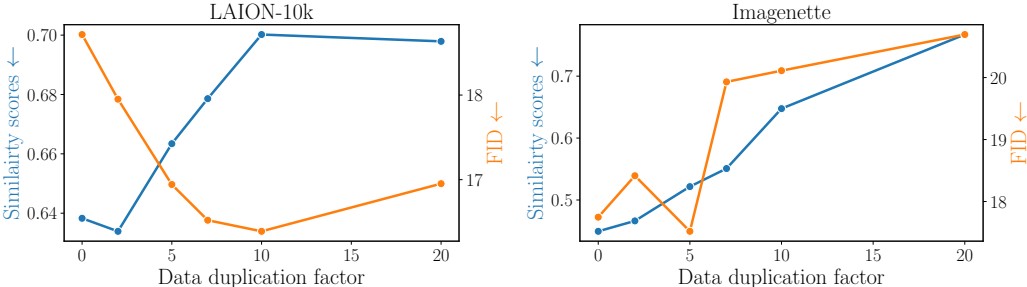

**Figure 3:** How does data duplication affect memorization? All models are trained with captions. On both datasets, dataset similarity increases proportionally to duplication in training data. FID score are unaffected by light duplication, but increase on higher levels as image diversity reduces.

data replication has been observed in newer models trained on de-duplicated data sets [Nichol, 2022, Mostaque, 2022]. Our goal here is to quantify the extent to which images are duplicated in the LAION dataset, and understand how this impacts replication at inference time. We will see that data duplication plays a role in replication, but it cannot explain much of the replication behavior we see.

## 5.1 How Much Duplication is Present in LAION?

In order to understand how much data duplication affects Stable Diffusion, we first identify clusters of duplicated images in LAION. We use 8M images from LAION-2B for our duplication analysis. Since duplicates are often duplicated many times, even a small subset of LAION is sufficient to identify them. Note that we do not use a random subset, but instead a consecutive slice of the metadata. Since the metadata is structured to provide URLs from similar domains together, this makes us likely to find large clusters.

We use SSCD [Pizzi et al., 2022] to compute the $8M \times 8M$ similarity matrix between all images within a slice of metadata. We threshold the similarity scores, only keeping ones above $0.7$ to identify images that are nearly identical. Then, we identify each connected component in the sparse similarity matrix and interpret it as representing a duplicated image. We show several of these clusters in the Appendix. We only consider clusters with at least 250 samples to ensure the images are duplicated enough times, which leaves $\sim 50$ clusters with a total of around 45K images.

To measure how similar captions are within a cluster, we first compute CLIP text features [Radford et al., 2021]. Then, we compute the similarity between captions using the dot product of CLIP text features multiplied by the unigram-Jaccard similarity between the captions of a cluster and finally compute the median value. We use the unigram-Jaccard similarity in addition to CLIP, since it better captures word-level semantics. We select $40$ captions from each cluster and feed them to Stable Diffusion. Finally, we determine if the generated images are replicas of their source cluster again using SSCD.

Results are shown in Figure 2. We observe that Stable Diffusion v1.4 shows high pixel-level replication (reflected in SSCD scores $> 0.4$) for some duplicated clusters, but only when caption similarity within the cluster is also high. In contrast, since the dataset for Stable Diffusion v2.1 was de-duplicated before training, it shows nearly no test-time replication for our identified clusters [Mostaque, 2022]. Despite being de-duplicated, Stable Diffusion v2.1 still exhibits memorization when scanning through a larger portion of the dataset, as we observed previously in Figure 1, showing that replication can happen even when the duplication rate in the training set is greatly reduced. In both cases, we see that image duplication is not a good predictor of test-time replication.

## 5.2 Controlled Experiments with Data Duplication

We train diffusion models with various levels of `data duplication factor (ddf)`, which represents the factor by which duplicate samples are more likely to be sampled during training. We train each model for 100k iterations and evaluate similarity and FID scores on 4000 generated samples.

Figure 3 contains results for LAION-10k and Imagenette. We observe that increased duplication in the training data tends to yield increased replication during inference. The relationship between data duplication and similarity scores is not straightforward for LAION-10k . As the duplication factor increases, similarity scores rise again until reaching a data duplication factor `ddf` of 10, after which they decrease. Regarding FID scores, we find that a certain level of data duplication contributes to

improving the scores for models trained on both datasets, possibly because FID is lowest when the dataset is perfectly memorized (if we infer with all training data captions).

**Other Observations from the Literature.** Somepalli et al. [2022] found that unconditional diffusion models can exhibit strong replication when datasets are small, despite these training sets containing *no* duplicated images. Clearly, *replication can happen in the absence of duplication*. As the training set sizes grow ($\sim 30$k), the replication behaviors seen in Somepalli et al. [2022] vanish, and dataset similarity drops, even when the number of epochs is kept constant. One might expect this for large enough datasets. However, models trained on the *much larger* LAION-2B dataset do exhibit replication, even in the case of Stable Diffusion 2.1 (Figure 1) which was trained on (at least partially) de-duplicated data. We will see below that the trend of replication in diffusion models depends strongly on additional factors, related especially to their *conditioning* and to *dataset complexity*.

> **Takeaway:** When training a diffusion model, it is crucial to carefully manage data duplication to strike a balance between reducing memorization and ensuring high-fidelity of generated images. However, controlling duplication is not enough to prevent replication of training data.

## 5.3 The Effect of Model conditioning

To understand the interplay between model conditioning and replication, we consider four types of text conditioning:

- **Fixed caption:** All data points are assigned the same caption, `An image`.
- **Class captions:** Images are assigned a class-wise caption using the template `An image of <class-name>`.
- **Blip/Original captions:** Each point is trained on the original caption from the dataset (LAION-10k ) or a new caption is generated for each image using BLIP [Li et al., 2022] (Imagenette).
- **Random captions:** A random sequence of 6 tokens is sampled from the vocabulary used to uniquely caption each image.

By varying caption scenarios, we transition from no diversity in captions ("fixed caption") case to completely unique captions with no meaningful overlap ("random caption") case. We again finetune on the Imagenette dataset, now using these caption types. As a baseline, we consider the pretrained Stable Diffusion model without any finetuning. Figure 4 (left) shows the dataset similarity among the baseline and models finetuned using the different caption types.

We observe that the finetuned models exhibit higher similarity scores compared to the baseline model. Furthermore, the level of model memorization is influenced by the type of text conditioning. The "fixed caption" models exhibit the lowest amount of memorization, while the "blip caption" models exhibit the highest. This indicates that the model is more likely to memorize images when the captions are more unique or have high perplexity. However, the model does not exhibit the highest level of memorization when using "random captions", meaning that captions should also be correlated with image content in order to help the model retrieve an image from its memory maximally.

**Training the text encoder.** So far, the text encoder has been frozen during finetuning. We can amplify the impact of conditioning on replication by training the text encoder during finetuning. In Fig. 4 (right), we observe a notable increase in similarity scores across all conditioning cases when the text encoder is trained. This increase is particularly prominent in the cases of "blip captioning" and "random captioning". These findings support our hypothesis that the model is more inclined to remember instances when the captions associated with them are highly specific or have high perplexity, or even unique, keys.

> **Takeaway:** Caption specificity is closely tied to data duplication. Highly specific (or high perplexity) captions act like keys into the memory of the diffusion model that retrieve specific data points.

### 5.3.1 The Impact of Caption vs. Image Duplication

In this section, we control separately for duplication of images and duplication of their captions to better understand how the two interact.

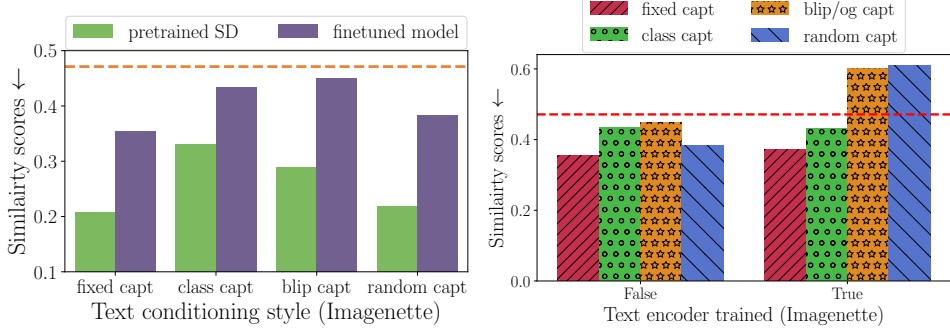

**Figure 4: Left:** Diffusion models finetuned on Imagenette with different styles of conditioning. FID scores of finetuned models are as follows (in order) 40.6, 47.4, 17.74, 39.8. **Right:** We show the effects of training the text encoder on similarity scores with different types of conditioning. Dashed lines are 95 percentile similarity scores of training data distribution.

In the case of **full duplication**, both the image and its caption are replicated multiple times in the training data. On the other hand, **partial duplication** involves duplicating the image multiple times while using different captions for each duplicate(although the captions may be semantically similar). To study the partial duplication scenario, we generate 20 captions for each image using the BLIP model for both Imagenette and LAION-10k datasets. For the full-duplication case in the LAION-10k experiments, we keep the original caption from the dataset.

We present the results on LAION-10k and Imagenette in Figure 5. We investigate how dataset similarity changes for both full and partial image-caption duplication at varying levels of duplication. Overall, our findings demonstrate that partial duplication consistently leads to lower levels of memorization compared to full duplication scenarios.

In Figure 5 (left), we compare the similarity scores of several models: the pretrained checkpoint, a model finetuned without any data duplication, a model finetuned with full duplication (`ddf=5`), and a model finetuned with partial duplication (`ddf=5`). We include dashed horizontal lines representing the background self-similarity computed between the dataset and itself. In the Imagenette case, models trained without duplication and with partial duplication exhibit dataset similarity below the baseline value, indicating a lower level of memorization. In contrast, the model trained with full duplication demonstrates higher levels of memorization compared to both the baseline and other cases. In the LAION-10k experiments, the model trained without duplication surpasses the training data similarity baseline. This observation raises the question of whether the memorization is inherited from the pretrained model, considering it is also trained on the larger LAION-2B dataset. However, when we compute the similarity scores on the pretrained checkpoint, we observe significantly lower values, indicating that the observed memorization is acquired during the fine-tuning process.

In Figure 5 (middle, right), we analyze how the similarity scores and FID scores vary at different `data duplication factors (ddf)` for full and partial data duplication. As the `ddf` increases, we observe an increase in memorization for models trained with full duplication. However, for partial

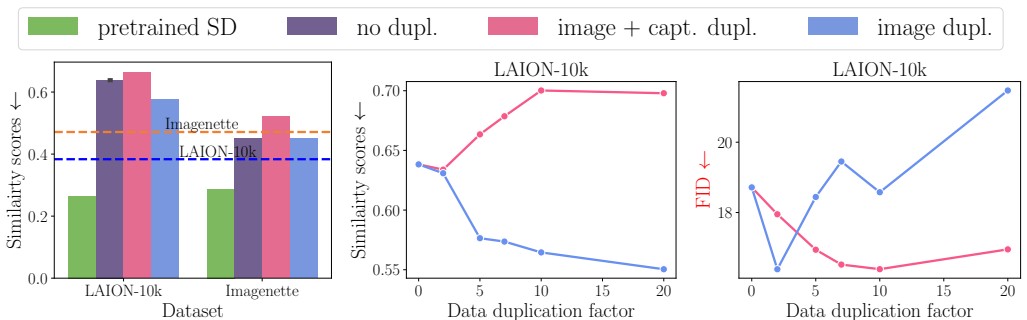

**Figure 5:** Models trained with different levels duplication and duplication settings. **Left:** Dataset similarity between models trained with no duplication, with partial duplication, and full duplication. Dashed lines show dataset similarity of each training distribution. **Middle, Right:** Dataset similarity and FID for full duplication vs partial duplication for different `data duplication factors`.

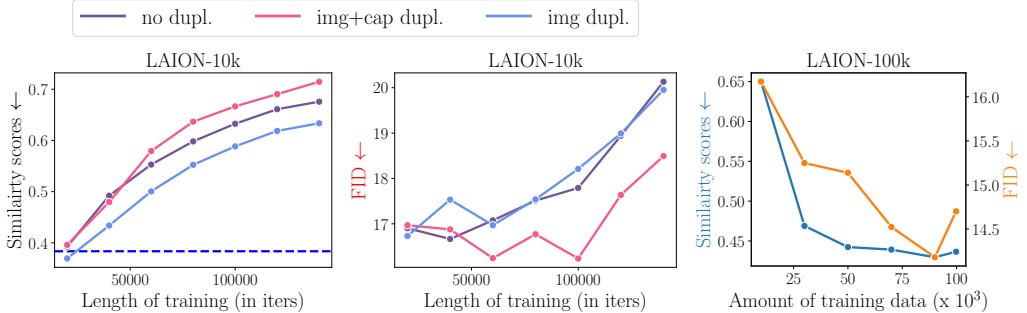

**Figure 6:** Does training for longer increase memorization? **Left, Middle:** Similarity and FID scores of models trained for different training number of iterations with different types of data duplication (`ddf=5`). **Right:** Metrics on models trained with different fractions of LAION-100k subset.

duplication, dataset similarity actually *decreases* with increased duplication. In our previous analogy, we now have multiple captions, i.e. keys for each duplicated image, which inhibits the memorization capabilities of the model. However, this memorization reduction comes at the cost of a moderate increase in FID at higher duplication levels.

> **Takeaway:** Compared to full duplication, partial duplication substantially mitigates copying behavior, even as duplication rates increase.

## 6   Effect of the Training Regimen

In this section, we examine how the training process and data complexity influence the degree of memorization.

**Length of training.**   Training for longer results in the model seeing each data point more times, and may mimic the impacts of duplication. In Figure 6 (left), we quantify the increase in dataset similarity at different training iterations. Notably, the image-only duplication model consistently exhibits lower similarity scores across epochs, while the image & caption duplication model achieves the highest values, showing that caption diversity for each duplicated image significantly slows down memorization. We show matching FID scores in Figure 6 (middle).

**Quantity of training data.**   Somepalli et al. [2022] demonstrate that diffusion models, specifically DDPM [Ho et al., 2020] models without text conditioning, exhibit a tendency to replicate the training data when trained on small datasets. In this study, we extend this analysis to text-conditioned models. To conduct this investigation, we now randomly sample a subset of $100,000$ images from LAION-2B. We then train using different fractions of this dataset, keeping other hyperparameters constant including the number of training iterations. The results of this experiment are found in Figure 6 (right). We observe that increasing the quantity of training data generally leads to lower similarity and FID scores. This aligns with the intuition that a larger and more diverse training set allows the model to generalize better and produce more varied outputs.

**Image Complexity.**   Throughout our experiments, we consistently observed a higher level of test-time replication in LAION-10k  models compared to Imagenette models, even when trained with the same levels of data duplication. We put forth the hypothesis that this discrepancy arises due to the inherent diversity present in the LAION-10k  dataset, not only in terms of images but also in terms of image complexity. To quantify image complexity, we employ two metrics. Histogram entropy measures entropy within the distribution of pixel intensities in an image, and JPEG compressibility (at JPEG quality 90), measures the size of images after compression. We resize and crop all images to the same resolution before computing these metrics.

Figure 7 (right) presents the distributions of entropy scores for LAION-10k  and Imagenette. LAION-10k  exhibits a wide range of data complexity, while Imagenette, predominantly comprised of complex real-world images, is characterized by higher complexity. Motivated by this observation, we explore the relationship between similarity scores and complexity of the top training point matched to each generation of a diffusion model trained on LAION-10k  without any duplication. Remarkably, we

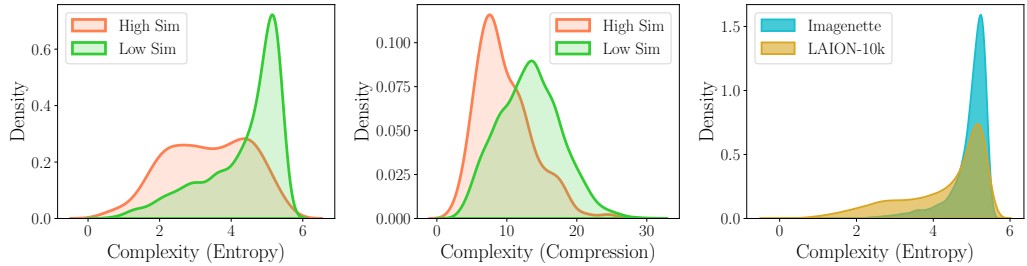

**Figure 7: Image complexity vs Memorization** We compute complexity scores using histogram entropy (left) and JPEG compressibility (middle) for memorized and non-memorized subpopulations, and compare both datasets (right). Refer to Sec. 6 for details.

discover a statistically significant correlation of $-0.32/-0.29$ (with p-values of $8e-98/7e-80$) for the entropy and compression metrics, respectively. We provide results for other duplication settings in the supplementary material, where we also find significant correlation. We present density plots illustrating the distributions of both complexity metrics for high similarity points (with scores $> 0.6$) versus low similarity in Figure 7 (left, middle). The clear separation between the distributions, observed across both complexity metrics, strongly suggests that diffusion models exhibit a propensity to memorize images when they are "simple".

> **Takeaway:** Choosing the optimal length of training, quantity of training data, and training setup is a trade-off between model quality and memorization. When images do get memorized in the absence of training data duplication, they are likely to be "simple" in structure.

## 7 Mitigation strategies

We have seen that model conditioning plays a major role in test-time replication, and that replication is often rare when image captions are not duplicated. Armed with this observation, we formulate strategies for mitigating data replication by randomizing conditional information. We study both training time mitigation strategies and inference time mitigation strategies. Training strategies are more effective, whereas inference-time mitigations are easily retrofitted into existing diffusion models. We experiment with randomizing the text input in the following ways:

- **Multiple captions (MC).** We use BLIP to make 20 captions for each image and we randomly sample all of them (plus the original) when training. This is for train time only.
- **Gaussian noise (GN).** Add a small amount of Gaussian noise to text embeddings.
- **Random caption replacement (RC).** Randomly replace the caption of an image with a random sequence of words.
- **Random token replacement & addition (RT).** Randomly replace tokens/words in the caption with a random word, or add a random word to the caption at a random location.
- **Caption word repetition (CWR).** Randomly choose a word from the given caption and insert it into a random location in the caption.
- **Random numbers addition (RNA).** Instead of adding a random word that might change the semantic meaning of the caption, we add a random number from the range $\{0, 10^6\}$.

In this section, we present the effectiveness of various mitigation strategies, summarized in Table 1. During train time, we find the multiple-caption strategy to be the most effective, which substantially increases caption diversity among duplicates. This matches our analysis in Fig. 5. At inference time strategies, we find that we can still inject a shift in the text prompt through random token addition/replacement. Such a shift again disrupts the memorized connection between caption and image. We verify based on FID and Clipscore [Hessel et al., 2021] that all mitigations have a minimal effect of model performance.

Images generated from models trained/tested with the best performing strategies can be found in Figure 8. For train mitigations we show our finetuned models, whereas for inference time we show mitigation strategies applied directly to Stable Diffusion. We show four images for each strategy, displaying the memorized image, generation from the regular model and from the mitigated model

**Table 1:** We present the similarity scores for models trained on LAION-10k with various train and inference time mitigation strategies discussed above. We show the results on no duplication, image duplication (ddf=5) and image & caption duplication (ddf=5) scenarios. For similarity scores, the lower the better. Best numbers with in train or test time are shown in bold. * collapses to the same setup.

| Dup style↓ / Mit strat.→ | None | MC | GN | RC | RT | CWR | GNI | RT | CWR | RNA |
|---|---|---|---|---|---|---|---|---|---|---|
| | | | Train time mitigation | | | | Inference time mitigation | | | |
| **No Dupl.** | 0.638 | **0.420** | 0.560 | 0.565 | 0.638 | 0.614 | 0.615 | **0.524** | 0.576 | 0.556 |
| **Image Dupl.** | 0.576 | **0.470\*** | 0.508 | 0.586 | 0.645 | 0.643 | 0.553 | **0.489** | 0.522 | 0.497 |
| **Image + Caption Dupl.** | 0.663 | **0.470\*** | 0.598 | 0.580 | 0.669 | 0.644 | 0.639 | **0.555** | 0.602 | 0.568 |

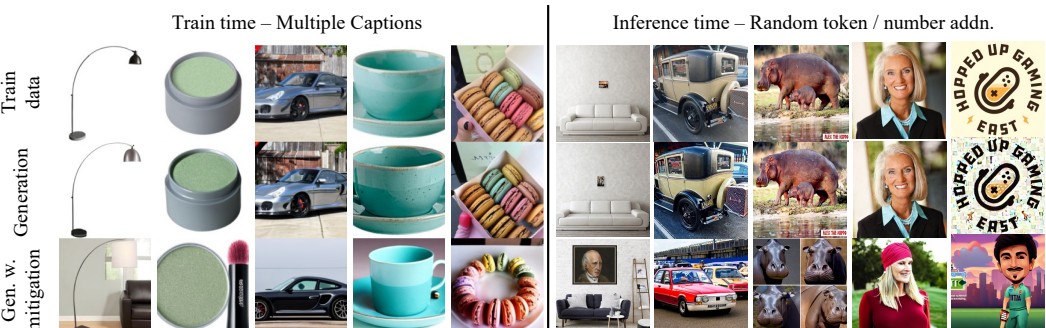

Train time – Multiple Captions      Inference time – Random token / number addn.

**Figure 8:** We present qualitative results on the best-performing train and inference time mitigation strategies. In each column, we present a potential training image model copied the generation from, a generation, and a generation on a model trained with a mitigation strategy/ mitigation applied to the original model. We used an image & caption duplication model with ddf=5 for the train time results and Stable Diffusion V1.4 (seed=2) to evaluate the inference-time strategies.

in each column. We fix the generation seed in each column. Overall, we find these mitigations to be quite effective in reducing copying behavior, even in large modern diffusion models. We refer the reader to the appendix regarding the hyperparameters used in the generation of Tab. 1 and the captions used for creating Fig. 8.

## 8   Recommendations for a safer model

Why do diffusion models memorize? In this work, we find that, in addition to the widely recognized importance of de-duplicating image data, conditioning and caption diversity also play fundamental roles. Moreover, we find in Fig. 5 that memorization can decrease even when duplication increases, as long as captions are sufficiently diverse. These findings help to explain why large text-conditional models like Stable Diffusion generate copies. Finally, we collect our observations into a series of recommendations to mitigate copying.

**Before training:** Identify large clusters of image data with significant object overlap, as measured by specialized copy detection models such as SSCD, and collapse them as described in Sec. 5.1. Likewise, clusters of caption overlap are potentially problematic, and duplicate captions can be resampled, for example using BLIP. In some cases, these clusters may need to be hand-curated because some concepts, like famous paintings or pictures of the Eiffel Tower, are acceptable duplications that the model may copy. Another avenue is to exclude images with limited complexity, see Sec. 6.

**During training:** We find training with partial duplications, where captions for duplicate images are resampled, to be most effective in reducing copying behavior, see Sec. 7. It may be worthwhile to use a low bar for duplication detection, as near-duplicates do not have to be removed entirely, only their captions resampled.

**After training:** Finally, after training, inference-time strategies can further reduce copying behavior, see Sec. 7. Such mitigations could be provided either as a user-facing toggle, allowing users to resample an image when desired, or as rejection sampling if a generated image is detected to be close to a known cluster of duplications or a match in the training set.

While text conditioning has also been identified as a crucial factor in dataset memorization for the diffusion model in our analysis, it is important to acknowledge that other factors may also influence the outcome in complex ways. Therefore, we recommend that readers thoroughly evaluate the

effectiveness of the proposed mitigation strategies within the context of their specific use cases before deploying them in production.

**Future Directions:** This research represents a first investigation into the influence of captions on memorization within diffusion models. We posit that examinations on broader scales, involving larger datasets and models, will further solidify our findings. Moreover, while our adopted similarity metric serves its purpose, we anticipate and encourage future studies to refine this metric or introduce alternatives grounded in a more robust theoretical framework on memorization.

# 9    Acknowledgements

This work was made possible by the ONR MURI program, DARPA GARD (HR00112020007), the Office of Naval Research (N000142112557), and the AFOSR MURI program. Commercial support was provided by Capital One Bank, the Amazon Research Award program, and Open Philanthropy. Further support was provided by the National Science Foundation (IIS-2212182), and by the NSF TRAILS Institute (2229885). Vasu Singla was supported in part by the National Science Foundation under grant number IIS-2213335.

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

# A Broader Impact

Text-to-image diffusion models are prone to violating intellectual property rights and causing harm to artists, even unbeknownst to the models' users. Given the rapidly increasing popularity of these models and also the likelihood that users will continue to generate and deploy their images whether or not copying behaviors are addressed, mitigating such behavior can prevent real-world harm. We take the first step towards mitigation by building up an understanding of how and why diffusion models copy their training data. Moreover, we show how to build text-to-image diffusion models which simultaneously produce high-quality images and at the same time do not replicate their training data nearly as often as popular existing models. Another important avenue for preventing copying is constructing detection pipelines so that we can identify copies of training samples before they are deployed, as in Somepalli et al. [2022].

# B User Caption Study

## B.1 Stable Diffusion User Matches False Positives

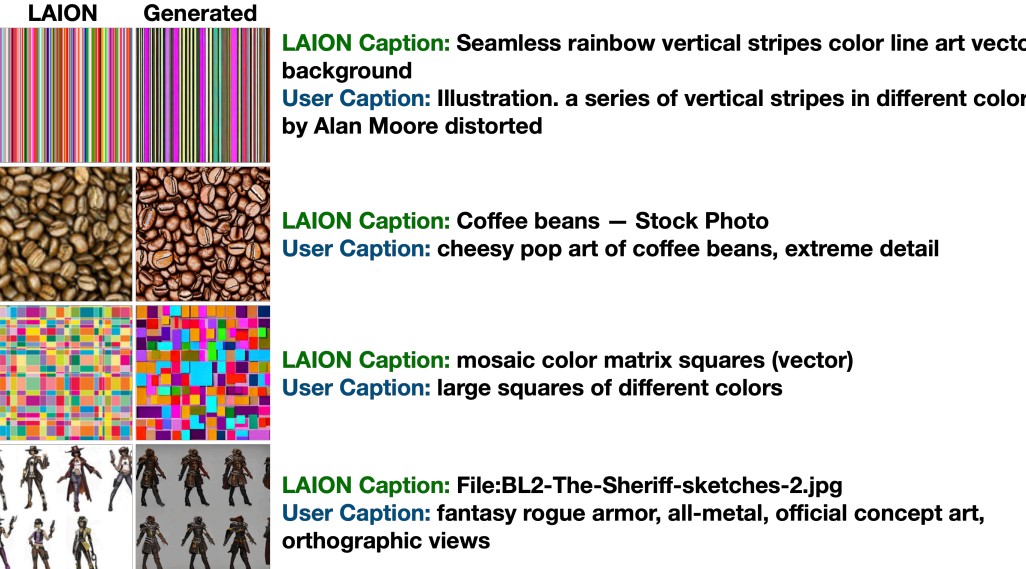

**LAION Caption:** Seamless rainbow vertical stripes color line art vector background
**User Caption:** Illustration. a series of vertical stripes in different colors. by Alan Moore distorted

**LAION Caption:** Coffee beans — Stock Photo
**User Caption:** cheesy pop art of coffee beans, extreme detail

**LAION Caption:** mosaic color matrix squares (vector)
**User Caption:** large squares of different colors

**LAION Caption:** File:BL2-The-Sheriff-sketches-2.jpg
**User Caption:** fantasy rogue armor, all-metal, official concept art, orthographic views

**Figure 9:** False Positive matches for User Captions using SSCD. Images are generated using Stable Diffusion v2.1, and corresponding closest matches from LAION subset are shown together.

In Figure 9, we show failure cases for SSCD match from the LAION subset to generated images from user captions. Several false positives are reasonable errors, caused by either high style similarity or simple texture of the generated images. A few of the errors are also caused for animated images as shown in the last row. We also observe a few cases where generated, and LAION images show significant domain shift from real images. This may be due to the training data of SSCD, which mostly consisted of real images Pizzi et al. [2022].

## B.2 User and LAION Captions for Images

The captions for the generated images using user prompts, and corresponding LAION matches for Figure 1 are shown below in order -

1. **User Caption -** "flower made of fire, black background,art by artgerm"
   **LAION Caption -** '"""""""Colorful Cattleya Orchids"""" (2020) by JessicaJenney"""'

2. **User Caption -** "the moon landing underwater "
   **LAION Caption -** "H/O: Buzz Aldrin on the Moon"

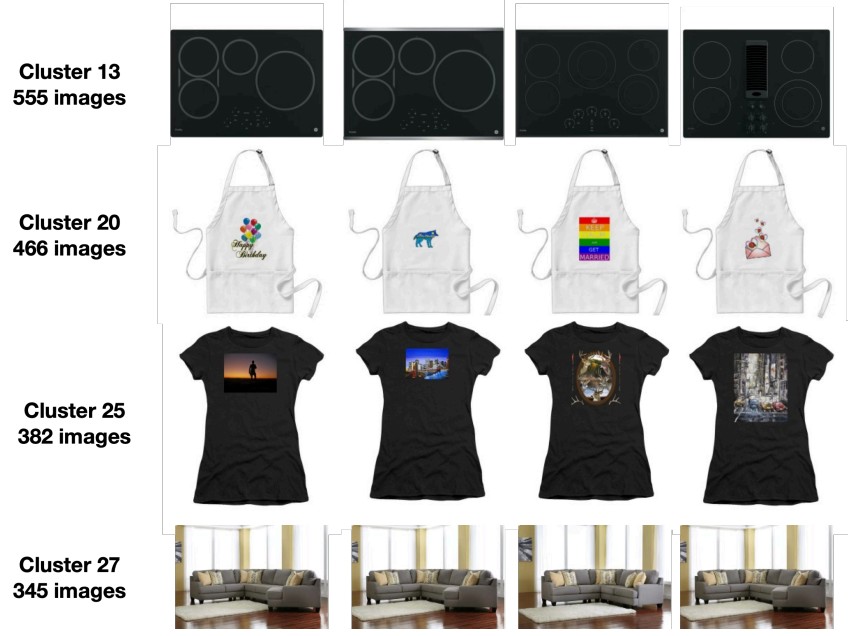

**Figure 10:** Clusters discovered from 8M subset of LAION using SSCD

3. **User Caption -** "rosie the riveter with a catfish head"
   **LAION Caption -** "We can do it! Canvas Print"

4. **User Caption -** "pubg splash art advertisement by masashi kishimoto / hiroya oku, hayao miyazaki, concept art, vibrant colors, 8 k "
   **LAION Caption -** "Minor kills brother for not letting him play PUBG game in Thane"

5. **User Caption -** "the super mario logo "
   **LAION Caption -** "Thumb super mario run logo"

6. **User Caption -** "a film still from joker ( 2 0 1 9 ) "
   **LAION Caption -** "Joaquin Phoenix as THE Joker in Comic Villain Casting"

7. **User Caption -** "cyberpunk tufting rug "
   **LAION Caption -** "Liquid Pattern Print Design 04 Area Rug"

8. **User Caption -** "pubg splash art advertisement by masashi kishimoto / hiroya oku, hayao miyazaki, concept art, vibrant colors, 8 k "
   **LAION Caption -** "Minor kills brother for not letting him play PUBG game in Thane"

## C   LAION Clusters

In Figure 10, we show a few of the clusters discovered using our proposed approach discussed in Section 5.1. As shown several of the images share high visual similarities with each other. The number of images discovered in each cluster is also shown. Note that the number is very likely an underestimate since we only use an $8M$ subset of the LAION dataset. A few of the clusters are also obtained due to broken links or similar placeholder images. A few clusters are also obtained for similar bar graphs or line charts. This shows LAION consists of several duplicates, which can be easily discovered using our approach.

## D   Extended experimental settings

Unless otherwise specified, all the models are trained for 100k iterations with a batch size of 16. We used Adam optimizer for training with $\beta_1 = 0.9$ and $\beta_2 = 0.999$ and `weight decay`$= 1e - 2$ We used one RTX-A6000 per model and it took about 24 hours to train. For inference, we used RTX-A5000 and it took approximately 5 hours to create enough generations to compute our metrics.

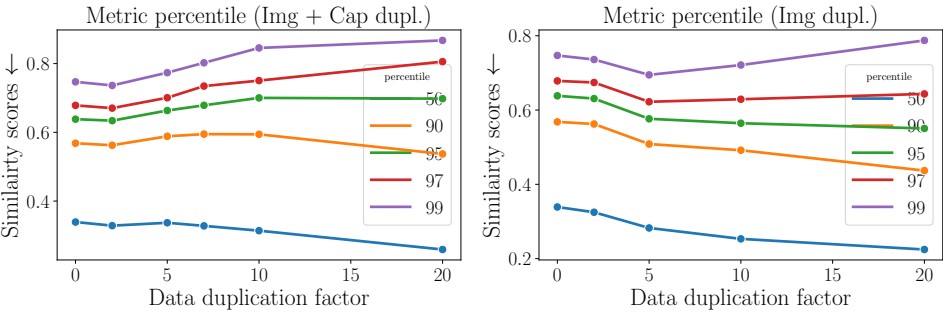

**Figure 11:** Similarity scores at different percentile cutoffs at different ddf models.**Left:** Full duplication **Right:** Partial duplication

To analyze various factors in this paper, we used approximately 5000 GPU hours for training and another 1000 GPU hours for inference.

# E    Discussion on similarity metric design

Occasional memorization, even in a small percentage of generations, can pose problems—precisely the pattern we've observed. To capture this behavior effectively, we measure the sim score of the top 95% of samples. Using the average or median, in contrast, could fail to detect significant memorization in a few instances. Our choice of the 95% percentile is based on observations by Somepalli et al., who identified around 2% of memorization. Adjusting for fine-tuning on a smaller dataset, we found 95% to be a suitable benchmark.

We have performed an ablation analysis on LAION-10k and full duplication across different data duplication factors and the results are presented in Fig. 11 (Left). We see 95, 97, and 99 percentiles follow similar trends. Median scores remain relatively constant, as expected, due to memorization affecting only a small percentage of images, as previously noted by Somepalli et al. Similar trends appear in partial duplication experiments, as evident in Fig. 11 (Right). Hence we proceeded with 95% value of similarity score distribution as the metric in this paper.

We present how similar or different the images look at different values of SSCD similarity scores in Fig. 12. We also present the similarity score histogram for the model trained with full duplication with ddf= 10 in Fig. 11 (Left). We also present a sample Precision-Recall curves on LAION-10k full duplication case in Fig. 11 (Right).

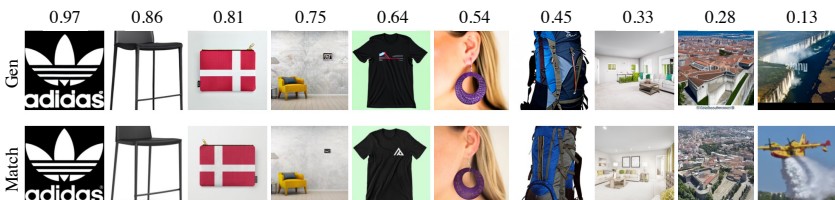

**Figure 12:** How similar the generation and the matches are at different SSCD scores.

# F    Why do diffusion models copy? Extended Results

In this section, we explore factors contributing to replication in diffusion models. We present additional results, including some figures from the main paper for completeness.

**Model conditioning.**    As shown in Fig. 4 (Left), higher uniqueness in conditioning captions leads to increased similarity scores, with Blip caption conditioning yielding the highest score. In Fig. 17 (Left), FID scores for fine-tuned models follow the same trends we have seen in the main paper, with the blip caption model exhibiting the lowest FID score.

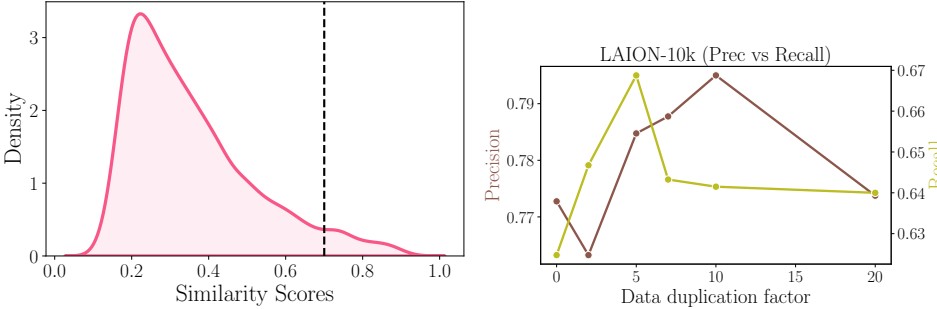

**Figure 13**

**Image Duplication vs Image + Caption Duplication** In Fig. 14, we analyze similarity and FID scores for models trained with varying duplication levels (`ddf`) on LAION-10k and Imagenette. Both datasets show increased similarity and FID scores with higher `ddf`. However, the impact of image duplication versus image + caption duplication differs. This disparity can be attributed to the diversity of captions used in image duplication for LAION-10k and the similarity of captions in Imagenette, potentially collapsing into image + caption duplication.

**Training Regimen - Length of Training** In Fig. 15, we examine similarity and FID scores during the training process on LAION-10k and Imagenette. We consider scenarios with no duplication, image duplication, and image + caption duplication. Longer training consistently leads to higher similarity and FID scores. However, image duplication consistently yields lower similarity scores compared to image + caption duplication on both datasets.

**Training Regimen - Text Encoder Training** In Fig. 4 (Right), we analyze the impact of text encoder training on similarity and FID scores for models trained with caption conditioning and various types of duplication on Imagenette and LAION-10k . Training the text encoder increases similarity scores while reducing FID scores in both datasets. In Fig. 17 (Right), we observe the effect of text encoder training on similarity scores for models trained with different conditioning types on Imagenette, specifically with image + caption duplication (`ddf=5`), and we see similar trends as of Fig. 4 (right) even in the presence of data duplication.

**Training Regimen - Image Complexity** In Fig. 18, we present distributions of self-similarity, entropy, and compression metrics for LAION-10k and Imagenette. LAION-10k exhibits lower self-similarity, indicating a distinct data distribution. However, complexity distributions are slightly shifted to the left, suggesting LAION-10k is marginally simpler than Imagenette.

Furthermore, we provide comprehensive findings on the correlation between similarity scores and training data complexity metrics in Tab. 2. We extensively evaluate models trained on LAION-10k without duplication, with image duplication, and with image + caption duplication, including cases with `ddf=5` and `ddf=20`. Notably, we consistently observe a statistically significant correlation in all scenarios. Intriguingly, the correlation is stronger when image + caption duplication is present compared to cases with only image duplication.

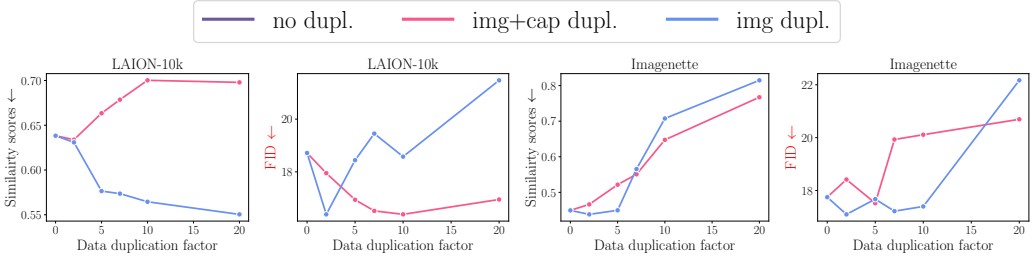

**Figure 14:** Diffusion models trained with different amounts of duplication and the style of duplication. We show similarity scores and FID scores at different values of `ddf` for LAION-10k and Imagenette models.

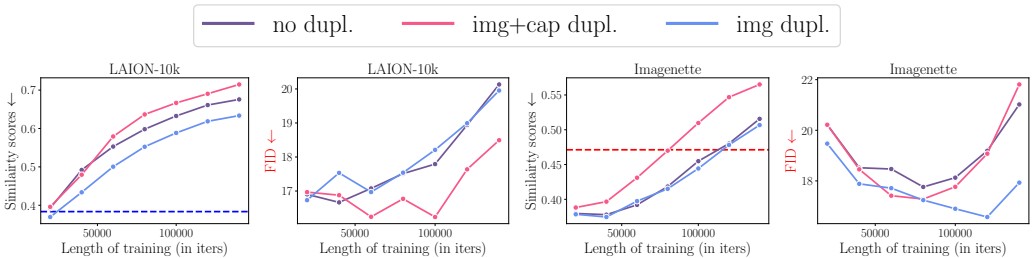

**Figure 15:** Does training for longer increase memorization?

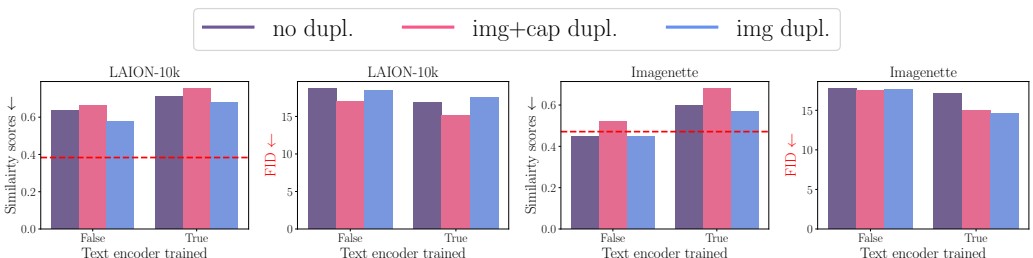

**Figure 16:** Effect of text encoder training on Similarity and FID scores.

**Table 2:** Correlation coefficients and corresponding p-values between the similarity scores and training data complexity metrics for different training scenarios.

| Duplication | ddf | Compression | Entropy | p-val (Comp.) | p-val (Entr.) |
|---|---|---|---|---|---|
| No duplication | 0 | -0.2926589 | -0.3230121 | 7.91E-80 | 8.48E-98 |
| Image duplication | 5 | -0.2298073 | -0.3034324 | 4.31E-49 | 5.80E-86 |
| Image + Cap duplication | 5 | -0.298469 | -0.3412334 | 4.19E-83 | 1.24E-109 |
| Image duplication | 20 | -0.0918793 | -0.1559931 | 5.83E-09 | 3.29E-23 |
| Image + Cap duplication | 20 | -0.1994873 | -0.2394177 | 3.50E-37 | 2.98E-53 |

# G   Misc. results.

We recreated the main paper Fig 2 using Roberta and presented it Fig. 19. The findings stay the same.

# H   Mitigation strategies

## H.1   Training details

Except for the affect of length of training on similairity scores experiment, we trained all the diffusion models for 100k iterations, including the models trained with a "mitigation" hueristic. The following are the additional hyperparameters relevent to a given strategy.

- **Multiple Captions (MC):** We sample 20 captions from BLIP for all training images. For LAION-10k , an additional original caption is included, resulting in a total of 21 captions.

- **Gaussian Noise (GN):** We sample random multivariate Gaussian noise $\sim \mathcal{N}(0, I)$, multiply it by a factor of 0.1, and add it to the latent CLIP representations of the caption.

- **Random Caption Replacement (RC):** With a probability of 0.4, we replace the caption of an image with a random sequence of words.

- **Random Token Replacement & Addition (RT):** With a probability of 0.1, we either replace tokens/words in the caption with a random word or add a random word at a random location. This step is performed twice for train time experiments and four times for inference time experiments.

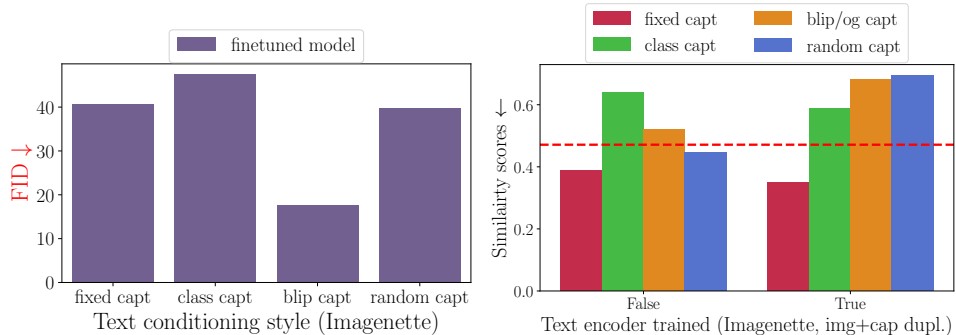

**Figure 17: Left:** FID scores of models trained on Imagenette with different types of conditioning. **Right:** Similarity scores of models trained on Imagenette with different conditioning and with and without text encoder training. All the models in this plot are trained with image + caption duplication with `ddf=5`.

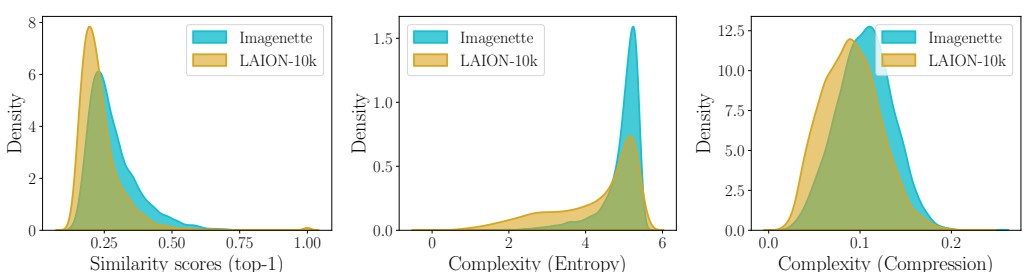

**Figure 18:** Dataset properties LAION-10k vs Imagenette. **Left:** Self-similarity **Middle:** Complexity - Entropy **Right:** Complexity - Compression

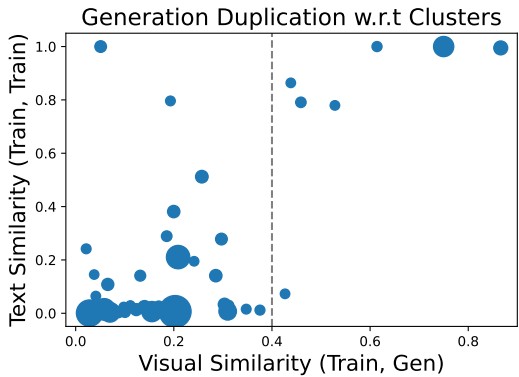

**Figure 19:** Replication of Figure 2 with Stable Diffusion v1.4 using RoBERTa for Text Similarity instead of CLIP

- **Caption Word Repetition (CWR):** We select a word from the given caption and insert it into a random location within the caption with a probability of 0.4. This step is performed twice for train time experiments and four times for inference time experiments.

- **Random Numbers Addition (RNA):** Instead of adding a random word that may alter the semantic meaning, we add a random number from the range $0, 10^6$ with a probability of 0.4. This step is performed twice for train time experiments and four times for inference time experiments.

## H.2  CLIP score and FID tables

In Tab. 1, we presented similarity scores for models trained/inferred with different mitigation strategies. In Tab. 4 and Tab. 3, we present the corresponding CLIP scores and FID scores. Gaussian Noise

strategy compromises on the FID score. Random Token/ Random Number addition during inference strategy compromises on CLIP score the most.

**Table 3:** FID scores of LAION-10k models trained with various mitigation strategies. Complementary to Tab. 1. *collapses to same case.

| Dup style↓ / Mit strat.→ | None | MC | Train time mitigation | | | | Inference time mitigation | | | |
|---|---|---|---|---|---|---|---|---|---|---|
| | | | GN | RC | RT | CWR | GNI | RT | CWR | RNA |
| **No Duplication** | 18.72 | 16.63 | 20.63 | 15.98 | 17.16 | 16.73 | 18.92 | 18.67 | 18.14 | 18.73 |
| **Image Duplication** | 18.44 | 16.22* | 21.62 | 15.92 | 15.32 | 16.03 | 19.04 | 19.23 | 18.38 | 19.19 |
| **Image + Caption Dupl.** | 16.94 | 16.22* | 18.34 | 15.63 | 16.25 | 16.26 | 17.36 | 17.75 | 16.84 | 17.35 |

**Table 4:** CLIP scores of LAION-10k models trained with various mitigation strategies. Complementary to Tab. 1. *collapses to the same case.

| Dup style↓ / Mit strat.→ | None | MC | Train time mitigation | | | | Inference time mitigation | | | |
|---|---|---|---|---|---|---|---|---|---|---|
| | | | GN | RC | RT | CWR | GNI | RT | CWR | RNA |
| **No Duplication** | 30.52 | 30.27 | 29.91 | 30.64 | 30.74 | 30.79 | 30.32 | 29.54 | 30.13 | 29.74 |
| **Image Duplication** | 30.27 | 30.03* | 29.52 | 30.54 | 30.86 | 30.83 | 29.96 | 29.10 | 29.74 | 29.22 |
| **Image + Caption Dupl.** | 30.62 | 30.03* | 30.30 | 30.57 | 30.76 | 30.79 | 30.42 | 29.69 | 30.13 | 29.74 |

## H.3 Extended Qual results

**Figure 20:** Applying Multiple Captions mitigation strategy on images memorized in SD 2.1

**Can we forget images memorized by SD 2.1 with MC strategy?** We used different images to show qualitative results for train and test time strategies due to the initial freezing of training data in our project. The image-caption pairs necessary for evaluating test-time mitigation using SD 2.1 were discovered later on.

However, to verify if train test strategies can make models forget the memorized images, we conducted an experiment by training two models, incorporating the additional 5 image-caption pairs (from Table 5 or Fig 8, right side). The first model is trained with ddf=5 and no mitigation, while the second model utilizes ddf=5 with the MultipleCaps (MC) mitigation strategy. The "train-time" results for these 5 images are presented in **??**. Encouragingly, with MC strategy, the model was able to forget 3 out of 5 images. However, the model still retained strong memorization for the prompts "Mothers influence on her young hippo" and "Classic Cars for Sale." We believe the need to combine multiple mitigations to address these remaining memorizations effectively.

**Inference time results.** The original and modified prompts used in creating figure Fig. 8 is shown in Tab. 5. For the generation, we used Stable Diffusion 1.4 with the generator `seed=2`, with default settings `guidance_scale=7.5`, `num_inference_steps=50`.

**Train time results.** In Fig. 21, we show extended train time qualitative results for all the mitigation strategies. We see "Multiple Captions (MC)" mitigation works **all** the time. In contrast, other methods might or might not always be effective.

**Table 5:** In first column, we show a few prompts that induce copying in SD 1.4 model. In the second column, we show the updated prompt after applying Random Token (RT) mitigation strategy. Sometime the random token is semantically related to the prompt and that might impact the Clipscore of the generation a bit more.

| Original Prompts | Modified Prompts |
|---|---|
| Wall View 003 | Wall disappointment View part senator 003 historian |
| Classic Cars for Sale | assen Classic dachsh compositions ;-) Cars for Sale |
| Mothers influence on her young hippo | Mothers 45460 influence on her 44791 young 32450 50192 hippo |
| Living in the Light with Ann Graham Lotz | Living in polaris the Light atrix with Ann Graham ancy Lotz turban |
| Hopped-Up Gaming: East | ita Hopped-Up Gaming: cricketer sati poutine East |

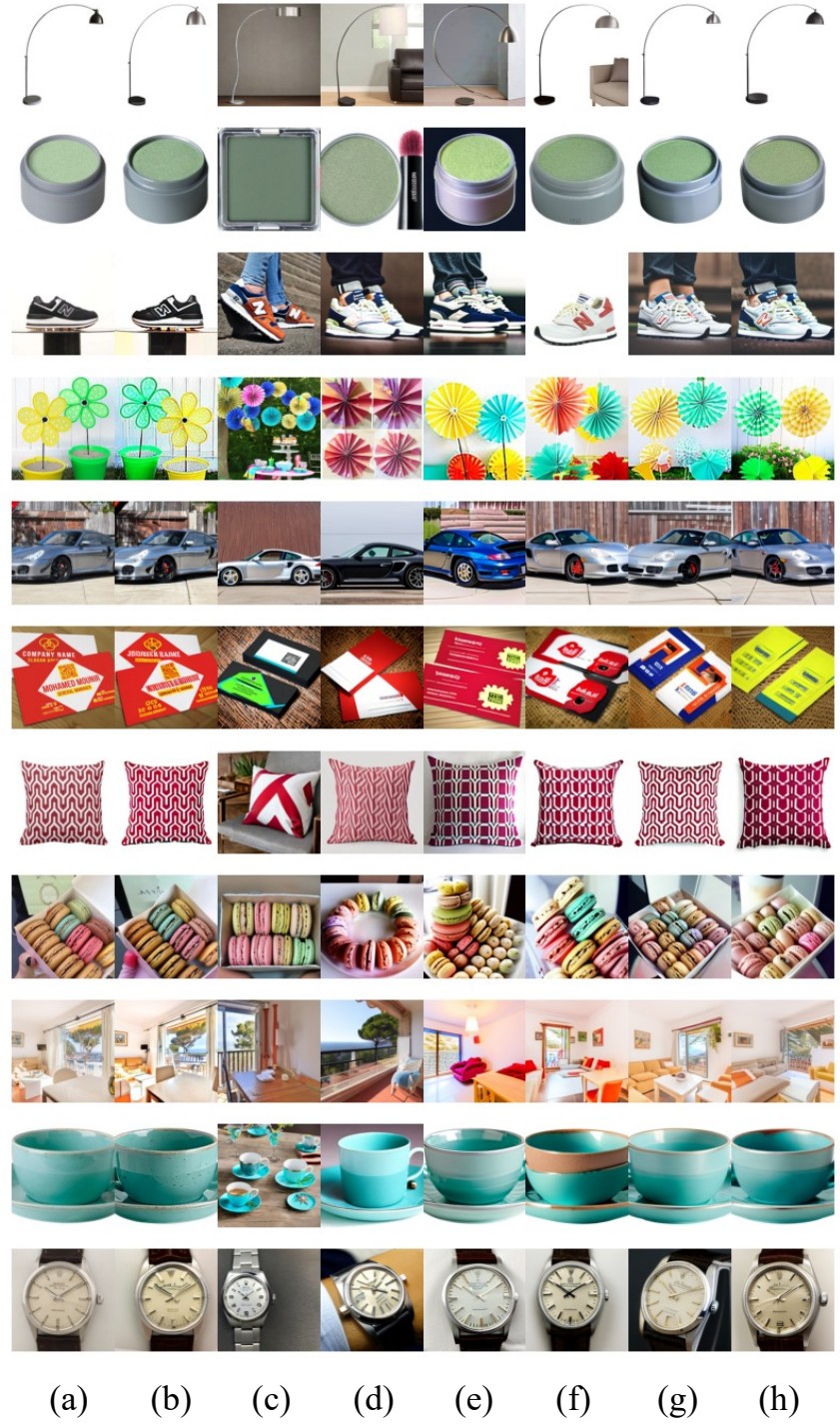

(a)   (b)   (c)   (d)   (e)   (f)   (g)   (h)

**Figure 21: Qualitative results of various train-time mitigation strategies** (a) Training image (b) Generated image without mitigation (c) SD model generation (d) Gen from MC strategy model (e) Gen from GN strategy model (f) Gen from RC strategy model (g) Gen from RT strategy model (h) Gen from CWR strategy model. Please refer to Appendix H.1 for the full forms of the mitigation strategies. MC strategy is consistently better.

