# OpenReview forum: "Understanding and Mitigating Copying in Diffusion Models"
_NeurIPS.cc/2023/Conference — NeurIPS 2023 poster_

### Official Review · Reviewer_R9Bm · 2023-07-05

**Soundness:** 3 good
**Presentation:** 4 excellent
**Contribution:** 4 excellent
**Rating:** 7
**Confidence:** 5

**Summary:**

This paper aligns with a range of recent works which investigate the tendency of diffusion models to memorize  training data and emit this during sampling, when prompted to, and presents strategies to mitigate this for data-processing, training and inference. Other than previous works, which mainly focused their analyses around duplicate images in the training data and model overfitting (i.e. dataset size, portion of train examples seen during training, …) , this work separately analyzes the influence of both these points and further adds to this the influence of captions (and their duplication) in the training data and complexity of the images in the training dataset. The main findings for these are that highly specific captions for every image promotes duplication which is furthermore also more likely for images which are simple (as measure by pixel histogram-entropies and jpeg-compressability).

**Strengths:**

* The paper separates the main factors of influence during training from each other and bases the findings made for each of these separate categories on rich sets of empirical experiments, making the paper conclusive and interesting
* The main take-aways are novel and of high importance for the community, especially since the authors center their experiments around fune-tuning a foundational DM, which is what many researchers do nowadays.
* The key take-aways for every analyzed influencing factor are summarized below the respective sections which makes the paper well structured and raises comprehensibility.

**Weaknesses:**

* It is totally reasonable that the authors cannot train a foundation model like SD from scratch and therefore focus on finetuning based on these models. However, the exact influence of the pretraining on all the results presented on conclusions taken remains unclear to me after the paper. It would therefore be interesting to also include results of models trained from scratch for all of the presented experiments, since it would provide insights about the generality of the statements made.
* Related to the above point, I think the title may promise a bit too much, since it would need experiments from scratch (for models such as SD) to choose that title. Maybe sth. Like “Why finetuned diffusion models memorize and …” would be better. What du you think?
* To me, it is counterintuitive that the model trained with no duplication has a higher dataset similarit score than the one with partial dup. in Fig. 5. If I didn’t miss sth, this is never discussed and is therefore confusing. It might also raise concerns about the significance of the results presented in this part which I think is crucial for the entire paper. Could the authors share their take on this?
* For a reader not knowing SSCD it may be hard to estimate, how reasonable the chosen newly introduced metrics related to SSCD are, since many of these are based on thresholds. Therefore, it would be great of the authors could add similarity scores to all visual examples showing found duplicates (especially in Fig. 1).
* In line 91 it is stated that FID would measure image quality and diversity. I don’t agree for diversity since there are many models with low FID scores which actually have very low diversity (many GANs, see e.g. [1,2]). I’d therefore recommend to either additionally include Precision and Recall Metrics [2] in the evaluation or remove diversity here.

[1] Sauer, A., Chitta, K., Müller, J., & Geiger, A. Projected GANs Converge Faster.
[2] T Kynkäänniemi, T Karras, S Laine, J Lehtinen, T Aila ,  Improved precision and recall metric for assessing generative models

**Questions:**

How diverse are BLIP captions? It would be interesting to do the same analysis than made for LAION captions in Sec. 5.1.

**Limitations:**

There is not dedicated limitations section, however, the limited scope of the experiments, which focus only on finetuning is mentioned in the paper. I’d recommend to add  a dedicated limitations section and once more state this. Also it would be interesting to see this analysis for other types of models such as AR models or GANS (although I know that no comparably powerful models to SD are publicly available as of now). The fact that this misses could also go into such a section

---

> ### Author Rebuttal · Authors · 2023-08-08
>
> We sincerely thank the reviewer for your insightful comments and positive feedback on our paper. Your recognition of our rigorous empirical experiments in dissecting training factors, coupled with the acknowledgment of the novelty and significance of our main takeaways, is truly encouraging.
>
> ### [Do fine-tuned models tell the whole story?]
>
> We concur with the reviewer that fine-tuning alone may not provide a complete picture. To assess this, we initially benchmarked the pretrained models on inference captions to evaluate the level of memorization. As shown by the green bars in Figure 4 and Figure 5 (left), similarity scores were found to be minimal, suggesting that fine-tuning might not significantly contribute to additional memorization.
>
> To further investigate, we attempted to train the U-Net from scratch (while keeping the encoder and decoder modules frozen). However, on this relatively small dataset, the trained models exhibited high FID scores, and the generated results were not satisfactory. Scaling to a much larger dataset for this setting is computationally prohibitive, considering the multitude of models we need to train for this paper.
>
> Despite these challenges, we acknowledge the importance of providing an ablation study. For the camera-ready version, we plan to include additional results using a slightly larger dataset (~1M samples), with either LDM or DDPM backbone (as explored in Somepalli et al.[1]), to offer a more comprehensive analysis.
>
> ### [Title update]
> We agree that the current title might be a bit too general currently while we are studying mainly T2I models, we will definitely update the title and parts of the introduction to reflect that.
> ### [How can partial dup be less than no dup? ]
>
> In experiments where the image is duplicated but the caption is not, we used a set of captions for each training instance of the duplicated images.  To do this, we use BLIP to sample random captions for duplicated images at train time.  At test time, we sample using the original LAION captions.  The test-time sampling uses different captions for duplicated images than the (Blip) captions used to train, and this shift makes the model less likely to replicate the training images.
>
> To dig deeper into this matter, we've added **response pdf Fig 18** with various sampling approaches in test time. The new curve reflects the same strategy (BLIP captions for duplicated images) employed during both training and testing. We see that this strategy results in slightly more memorization than sampling with LAION captions, but still far less memorization than the case where both image and caption are duplicated during training.
>
> Even with these updated captions, for lower ddfs, the similarity scores are lower compared to no duplication case. We hypothesize this slight dip is caused by the model not memorizing "easy" images since some of the easy images are not sampled as much as in the case of no duplication since we train all the models to a fixed number of iterations. We base this hypothesis on manual observations made on the top 100 inferences with high similarity scores.
>
> ### [ How does the sim score and different SSCD thresholds look like?]
>
> We agree with the reviewer that adding SSCD similarities in Fig 1 could help the reader understand the content better, and we will update this in the camera-ready version. However, for the sake of your understanding, these are the SSCD numbers we observed in Figure 1, left to right 0.63, 0.60, 0.57, 0.57, 0.56, 0.58, 0.52.
>
> Please refer to **Fig 22 in response pdf** to see how much the images differ at different similarity scores. We also observed that similarity scores > 0.5 tend to be very close matches.
>
> ### [ L91 FID would measure image quality and diversity]
>
> We sincerely thank the reviewer for bringing this error to our attention. We will update the draft. In response to your suggestion, we have computed Precision and Recall based on the approach presented in Kynkäänniemi et al. [2]. In **Fig. 20 (right) of the response pdf**, we have included these metrics for the full duplication LAION-10k experiments. We firmly believe that incorporating these metrics will enhance the depth of our analysis. Therefore, we commit to including them for all LAION-10k experiments in the supplementary material of the camera-ready version.
>
> ### [How diverse are BLIP captions?]
> We thank the reviewer for the suggestion. We will add a similar analysis to Sec 5.1 using BLIP captions to the appendix in the final draft.  We perform preliminary exploration to understand the diversity of BLIP captions. We used the CLIP text encoder, to compute the cosine similarity for each image across BLIP captions, and observe median similarity b/w BLIP captions is $0.79 \pm 0.08$. The similarity of LAION captions to BLIP captions for the same images is $0.64 \pm 0.07$. Unfortunately, since LAION doesn't provide multiple captions per image we cannot use this metric to measure diversity for real captions.
>
> ### [Adding a limitations section]
>
> We appreciate the reviewer's valuable feedback regarding the scope of our experiments and the need for a dedicated limitations section. It is indeed a great suggestion to provide a comprehensive overview of the scope and assumptions underlying our experiments. We will update the conclusion to include the limitations in the final draft.
>
> Thank you for your thoughtful review. We hope we addressed your concerns/questions sufficiently and we would appreciate it if you would consider raising your score in light of our response. Please let us know if you have additional questions we can address.
>
> ---
> References:
>
> [1] - Somepalli, Gowthami, et al. "Diffusion art or digital forgery? investigating data replication in diffusion models." _Proceedings of the IEEE/CVF Conference on Computer Vision and Pattern Recognition_. 2023.
> [2] - T Kynkäänniemi, T Karras, S Laine, J Lehtinen, T Aila , Improved precision and recall metric for assessing generative models

---

> > ### Comment · Reviewer_R9Bm · 2023-08-18
> > **Thanks for the clarifications**
> >
> > Thanks for your answers, keeping my initial score.

---

### Official Review · Reviewer_MEzn · 2023-07-06

**Soundness:** 2 fair
**Presentation:** 3 good
**Contribution:** 3 good
**Rating:** 6
**Confidence:** 1

**Summary:**

This paper discusses the issue of data replication in text-to-image diffusion models, specifically focusing on the memorization problem. It mentions that these models, such as Stable Diffusion, are known for unintentionally replicating their training data, which has led to recent lawsuits. The commonly held belief is that duplicated images in the training set contribute to content replication during inference. However, the paper's analysis reveals that the text conditioning of the model also plays a significant role in data replication. The experiments show that unconditional models don't replicate data as frequently as text-conditional models. Building on these findings, the paper proposes several techniques to mitigate data replication during inference by randomizing and augmenting image captions in the training set.






**Strengths:**

- The paper is well-organized and well-written
- The experiment setup is clear and sound.
- The originality of resolving the memorization issue of the diffusion model is interesting.
- The experiment results are convincing.

**Weaknesses:**

- Even though this paper is focusing on the empirical results and evidence of memorization of the diffusion model, I am still expecting a high level of theoretical understanding. Specifically, does it holds for the general case regardless of the type of SDE, data distribution etc.?
I think these questions are also important.

**Questions:**

- It may be interesting to show some toy cases in 2 dimensional or 1 dimension in which one can simply play around with how the conditioning and replication will influence the vector field (probabilistic flow) of the generation. It might be a good justification of authors' hypothesis.

**Limitations:**

The observation is based on the heuristic observation, however, it is fine since it is impossible to have a white-box explanation of a neural network. It would be better to have some guidance in theoretical reasoning.

---

> ### Author Rebuttal · Authors · 2023-08-08
>
> We thank the reviewer for recognizing the merit of our work. We are pleased to know that you found our experiment setup clear and sound.
>
> ### [Toy cases]
> We thank the reviewer for this suggestion. In our camera-ready version, we will visualize a toy problem in 2 dimensions where we know the closed-form density function of the underlying data distribution precisely. In this case, we can visualize the deviation of the vector fields generated by diffusion models trained using various protocols, including our interventions, from the gradients of the true log-density.
>
> ### [Theoretical understanding]
>
> Understanding why diffusion models memorize, and why they even work well in the first place, is an exciting and active area for future theory research. Text-to-image models considered in this work are a form of classifier-free guidance models proposed by Ho et al. [1].  Existing research [1,2,3] has made limited progress towards building a theoretical understanding of diffusion models generally, and memorization remains unexplored as of yet.  Alternatively, [4] provided a framework for studying memorization in discriminative models, and future work may build up corresponding theories for diffusion models. In this work, we present an empirical investigation into factors which contribute to memorization in diffusion models with a focus on providing recommendations to training future diffusion models. We hope subsequent work will bridge the gap between theoretical underpinnings and empirical observations of memorization in diffusion models. We will update the discussion to underscore this as a promising trajectory for future research.
>
>
> Thank you again for your thoughtful review. We hope we addressed your concerns/questions sufficiently and we would appreciate it if you would consider raising your score in light of our response. Please let us know if you have additional questions we can address.
>
> -----
>
> References:
>
> [1] - Ho, Jonathan, and Tim Salimans. "Classifier-free diffusion guidance." arXiv preprint arXiv:2207.12598 (2022).
>
> [2] - Song, Yang, et al. "Score-based generative modeling through stochastic differential equations." arXiv preprint arXiv:2011.13456 (2020).
>
> [3] - Chen, Sitan, et al. "Sampling is as easy as learning the score: theory for diffusion models with minimal data assumptions." arXiv preprint arXiv:2209.11215 (2022).
>
> [4] - Feldman, Vitaly, and Chiyuan Zhang. "What neural networks memorize and why: Discovering the long tail via influence estimation." Advances in Neural Information Processing Systems 33 (2020): 2881-2891.

---

> > ### Comment · Reviewer_MEzn · 2023-08-16
> > **Response to authors**
> >
> > Thank you for your answer. I will keep the current score.

---

### Official Review · Reviewer_4RBY · 2023-07-07

**Soundness:** 2 fair
**Presentation:** 3 good
**Contribution:** 3 good
**Rating:** 4
**Confidence:** 4

**Summary:**

The paper tries to explain the memorization and copying of training data in diffusion models. The paper establishes a connection between caption duplications and memorization on top of data duplications analyzed in previous works. In their main experiments, the authors fine-tune various SD models on ImageNette/LAION subsets and show that controlling data duplication does not completely prevent memorization and that highly specific captions amplify copying. They also relate memorization to training parameters and data complexity. Finally, the paper introduces several mitigation strategies based on randomizing conditional information.


**Strengths:**

 - Overall, I find the experiment design in the paper to be good and got a better understanding of the memorization phenomenon from the paper. I also enjoyed the writing style of the paper.
 - I find the connection between image complexity and copying to be reasonable and believe this is the first time this has been analyzed this way.
 - While previous papers were mostly about analyzing the impact of data duplication on memorization, it makes sense to also control the captions and the paper shows that there exists a connection between the two.
 - The problem setting of the paper is very relevant.


**Weaknesses:**

 - Pretty much all experiments are based on the dataset similarity score which is defined as the 95-percentile of image-level similarities. As this is very important for the validity of the experiments, I believe that it would make a lot of sense to motivate this design choice further. From previous works, the authors quote that an image similarity larger than 0.5 shows strong similarities. In the worst case, the distribution of image similarities could contain slightly less than 5% near duplicates and slightly more than 95% unrelated images which would result in an arbitrarily small dataset similarity but still a decent memorization quote. Maybe the authors could simply demonstrate that the distribution of image similarities is well-behaved and also quantify the number of outliers, eg at a dataset similarity of 0.2, how many of the remaining images above the 95%-quantile actually achieve an image similarity larger than 0.5. Ideally, it would also be great to shortly demonstrate that results look similar with larger percentiles and that the 95-percentile is not critical for the outcome of the experiments.
- The range of similarity scores in most of the experiments is actually somewhat limited. I do not have a good feeling about the scale of this metric, but for example, the data duplication factor seems to only influence the score in the range 0.64 to 0.70. As this is the 95% quantile and 0.5 can be assumed to be a duplicate, this means that in all cases we have strong duplicates. I do believe the authors that there is a connection between duplicates/captions and copying, however, either the dataset score is not the right metric for this kind of experiment or duplicates/caption can actually only explain a relatively small part of the problem. Also even with the best-performing train time mitigation strategies, the similarity score is 0.42/0.47 so from my understanding this could mean that we still have about 5% of duplicates.
- The mitigation strategies are somewhat simplistic. I also believe that it is crucial to give a more detailed analysis of the "minimal effect" that the mitigations have on model performance. The images on the left half of Figure 8 work well in the sense that they seem to produce the same object that was also on the original image. However, most of the images on the right side of Figure 8 are not convincing to me. For example, I do not see any interaction between the hippo mother and her child (prompt: "Mothers influence on her young hippo"), and from a short Google search I would say that the person in the image is not Ann Graham Lotz (prompt: "Living in the Light with Ann Graham Lotz") and I could not find any connection between the last image with mitigation and "hopped up gaming east". I think mitigating copying is clearly important, but if this prevents the SD model from following the user's prompt it becomes useless. I think Figure 8 should show the results of the best train and test time mitigation strategies for the same images and I would also like to see the results of the best train time defenses on the images on the right side as apparently they seem to be harder. For some images in Figure 16, I also do not find any of the test time defenses to produce reasonable results.


**Questions:**

 - There are pretty strong differences between Imagenette and Laion-subsets, do you have also looked at other datasets and do they fall between the two datasets shown in the paper or do the results look completely different there?

**Limitations:**

In my opinion, one of the largest limitations is that even the best-performing training guidelines/mitigation strategies might not always prevent copying. The paper gives us a dataset similarity result of 0.42/0.47 and claims that in terms of image similarity, 0.50 would be a duplicate. So it is not clear to me that this actually solves copying.

---

> ### Author Rebuttal · Authors · 2023-08-08
>
> We thank the reviewer for their thorough feedback. We greatly appreciate your acknowledgment of our experiment design, improved comprehension of the memorization phenomenon, and writing style, and that you found our approach to controlling captions and image complexity analysis insightful.
>
> ### [Design choice of 95 percentile]
>
> We kindly ask the reviewer to consult the global response for our detailed response to this specific comment.
>
> ### [Understanding the range of similarity score metric]
>
> To understand what it means to go from simscore of 0.64 to 0.7, we looked at the inferences from different perspectives. We computed the number of points with high similarity scores for the model with no duplication and the model with duplication ddf=20. We found that the latter model has **2.5x** the number of points than the former model. It shows that the simscore does indeed reflect an amount of memorization in the model.
>
> We've further added an additional plot to explain how this metric is scaled. For example, if we duplicated 20%  of the training data. In **response pdf Fig 19**, we compare the results of 5% and 20% data duplication. This extends the range of the metric to 0.75 for the 20% duplication case.
>
>
> ### [Is duplication or caption the whole story?]
>
> To answer your question on whether copies exist in mitigated models, we computed % of images with high SSCD scores (>=0.5) for different models.  We present the numbers against similarity scores in the table below. (MC= multicaps mitigation)
>
> |  Model setting| Sim.Score | % of points > 0.5|
> | --| -- |--|
> |Nodup | 0.638| 17.65 |
> |Nodup + MC| 0.42| 2.17 |
> | Full dup, ddf=5 | 0.663 | 19.55|
> | Full dup, ddf=5 + MC | 0.47 | 3.85|
>
> While duplication and caption specificity explain a part of memorization, we believe that other factors, such as image complexity, may also play a role in influencing the extent of memorization. Notably, even with our most effective mitigation strategy, the model memorizes approx 2% of images, even when they were not duplicated during training. Qualitatively, they are simple images (e-commerce visuals) or brand logos (like Adidas).
>
>
> Our primary focus in this study was to address memorization arising from the caption side and successfully mitigate its impact. While we briefly explored the influence of image complexity on memorization, we acknowledge that there are more factors at play. We view these findings as a stepping stone, and we encourage future research to delve deeper into other contributing factors for a comprehensive understanding of memorization in diffusion models.
>
>
>
> ### [Reg. mitigation strategies]
>
> **How are mitigation strategies affecting the model performance?**
>
> We agree with the reviewer that test time strategies can impact the text guidance, we actually studied this impact quantitatively by comparing Clip Scores in Appendix Table 4. We see that the *impact is minimal* under many strategies. We also study how the FID scores are impacted when each of the mitigations is applied in Appendix Table 3. The increase in FID scores are also minimal for many mitigation strategies.
>
> Regarding the example, "Living in the Light with Ann Graham Lotz”, this image was observed as an undesirable memorization in a recent Google work [1]. This is, of course, subjective, we might still want models that generate people of public interest with high fidelity. Yet, we, as well as [1] are concerned with copyright problems, as the generated image is the exact copyrighted image corresponding to her radio program. Based on our definitions of memorization,  would rather not generate an image in the likeness of "Ann Graham Lotz", than generate a copyrighted image. We agree that there is a trade-off here, between fine-grained prompt following and memorization, and our work presents one possible choice. We will extend the discussion of the scope of memorization in our work to better reflect this.
>
>
> [1] - Carlini, Nicholas, et al. "Extracting training data from diffusion models." arXiv preprint arXiv:2301.13188 (2023).
>
> **Why we chose different samples for train, test time results?**
>
> We used different images to show qualitative results for train and test time strategies due to the initial freezing of training data in our project. The image-caption pairs necessary for evaluating test-time mitigation using SD 2.1 were discovered later on.
>
> In response to the reviewer's concerns, we trained two models, incorporating the additional 5 image-caption pairs (from Table 5 or Fig 8, right side). The first model is trained with ddf=5 and no mitigation, while the second model utilized ddf=5 with the MultipleCaps (MC) mitigation strategy. The "train-time" results for these 5 images are presented in **response pdf Fig. 23**. Encouragingly, with MC strategy, the model was able to forget 3 out of 5 images. However, it still retained strong memorization for the prompts "Mothers influence on her young hippo” and "Classic Cars for Sale." We believe the need to combine multiple mitigations to address these remaining memorizations effectively.
>
> **Multiple Captions strategy is consistently better.**
>
> We would like to point out that Fig 16 is all train time strategies. We apologize for being unclear and we will update the caption.  MultipleCaption (Fig 16, col c) mitigation consistently removed memorization in all of the cases. Although the other strategies may not have exhibited the same level of efficiency, we deemed it essential to present both the qualitative and quantitative results for all strategies. Our intent is to inspire and guide future research in this crucial area.
>
>
> Overall, we appreciate the reviewer's insights, and these additional experiments have added depth to our analysis. We hope we addressed your concerns/questions sufficiently and we would appreciate it if you would consider raising your score in light of our response. Please let us know if you have additional questions we can address.

---

### Official Review · Reviewer_rsTQ · 2023-07-13

**Soundness:** 2 fair
**Presentation:** 2 fair
**Contribution:** 4 excellent
**Rating:** 5
**Confidence:** 5

**Summary:**

This paper is a deep dive into the phenomenon of memorization in Diffusion models. I find that the paper is very well position given the current state of generative ML and the paper does a great job at asking the right questions about various factors concerning the memorization of images. Some of the key takeaways in the paper are really strong, such as randomizing captions, or partial duplication of images, but some others are not very well substantiated. Finally, the authors also conclude with some guidelines and mitigation strategies for practitioners, making this a refreshing read.

This paper is hard to review in a strengths/weakness pattern because each section has its strengths and weaknesses. First, this paper tries to do a lot. I think that ends up being a weakness of the paper because the high-level message from the abstract and intro gets swayed away through the body making this a very confusing read. Especially in the sections from 3 to 5.3. It is only after that the main narrative of the work catches on. I understand that the authors want to show a lot of experiments and analyses they did, but many of them are not at all central to the key message of the paper.


**Strengths:**

1. This paper tries to break down various factors that are responsible for the memorization of training data by diffusion models.
2. The discussions in sections 6 and 7 are really valuable to the community.
3. The paper asks the right questions and also performs experiments to support their hypothesis (more about this in the weaknesses). But overall, the paper serves as a great starting point for understanding the state of memorization in generative models, giving a peak at some thought-provoking questions, and using some of these hypotheses to actually suggest mitigation strategies.
4. I am curious about the image complexity argument. This is a refreshing take, but also very different from image classification literature where typically hard images are memorized because the model can easily learn the simple ones to classify. It seems that the take for generative models is the opposite. but again there is a distinction between anecdotal arguments of features versus the argument of compressibility and entropy studies here. A good metric potentially would be the embedding similarity as assessed by a CLIP model.

**Weaknesses:**

My biggest qualm with this paper is about the metrics. It seems that this paper is based in a delusional world where dataset similarity and FID are the sole indicators of model memorization and model performance. I know that evaluating both generation and memorization are hard problems, but this paper takes these yardsticks way too far, to an extent that following up with some results becomes uncomfortable. And in many sections, I get the sense that hypotheses and conjectures are drawn just to satisfy these so-called oracle notions of memorization and quality.

1. On generation quality: Section 7 severely needs an analsis on the impact of these changes on the quality of generation. Similary, in Figure 4, FID scores are all over the place. are these even good models?
2. On memorization: In Section 5.1 the authors note that they did not observe any duplication in the clusters they studies. Again telling how the metrics of DS are not informative or reflective of memorization in the first place. I do not understand why checking similarity scores with the top 5% of generations. Further, in Figure 5: lower SS when the image is duplicated. and this happens even in the right figure. But does this actually related to "memorization" or just stay till the idea of dataset similarity. For instance, in the de-duplication done for SD2, not all duplicates were "full duplicates" if I am correct. So i think the explanation saying random captions helps reduce SS is an over statement.
3. On section 3 and Figure 1 proper attribution should be given to the original authors who devised the technique
4. I have not seen people use clip text features to assess how close texts are. I would suggest using some text model such as RoBERTa for these use cases.
5. "possibly because FID is lowest when the dataset is perfectly memorized" I do not see how this logically flows. And in general I did not like that this paper makes too many conjectures to explain every possible phenomenon they see in results. It just feels like the authors are trying to fit a jigsaw no matter what.






**Questions:**

1. " Since the metadata is structured to provide URLs from similar domains together, this makes us likely to find large clusters." Do you have a reference to this? I do not think this is the case from my observations of similar images across various parquets in the LAION dataset
2. Are the results in 5.2 on unconditional models?
3. Figure 4: why does pretrained SD have different similarity scores?



**Limitations:**

-

---

> ### Author Rebuttal · Authors · 2023-08-09
>
> We are happy to learn that you perceive our contributions as both innovative and crucial in initiating a deeper understanding of memorization.
>
> ### [Generation quality]
> We provide FID scores in Appendix Table 3 for different mitigation strategies. Irrespective of data duplication, Multiple Captions strategy yields lower FID scores than no mitigation case. We agree with the reviewer that the FID scores in Fig. 4 bounce around a lot.  When we reviewed the generations of these models, they seem to be of reasonable quality (for models trained on only 10k images). We present a few images in **response pdf Fig 21**. It seems that the increases in FID are because of changes in the image distribution (as in the model is not learning all the classes equally well), and not in image quality.
>
> ### [Memorization]
> **Are DS similarity scores indicative of memorization?** Kindly refer to **"Is SSCD good for understanding memorization?"** section in the global response.
>
> **Not finding clusters in sec 5.1** We do not agree with the reviewer's characterization that we did not find meaningful clusters in SD 2.1.  While we found fewer & smaller clusters (likely because of de-duplication of the training set) in  SD 2.1 than we found in SD 1.4, clusters were still found, and an examination of these clusters revealed copying behavior (e.g., the "hopped up gaming'' cluster). Note that for SD v2.1, the dataset was explicitly cleaned of duplicate images using the PHash metric  (based on personal correspondence with the authors).
>
> While acknowledging the metric's limitations, our focus in section 5.1 is to demonstrate memorization under high text and visual similarity and perform a comparative SD v1.4 vs. SD v2.1 analysis.
>
> **Why top 95pc?** Kindly refer to **"The choice of percentile in SS metric"** section in the global response.
>
> **Why SS lower when the image is duplicated?** In partial duplication experiments, we used BLIP captions for duplicated samples and LAION captions for others. During inference, all captions are from LAION. We believe this shift slightly reduced replication. In **response pdf Fig 18**, SS with varied sampling approaches in test time is presented. While the strategy (BLIP captions for duplicated images) in training and testing slightly increases memorization compared to LAION captions, it is still less than image + caption duplication models.
>
> Even in the updated plot, a slight dip at lower ddfs remains. This might be due to the model not memorizing "easy" images as non-duplicated easy images are sampled less. We base this hypothesis on manual observations made on the top 100 inferences with high similarity scores.
>
> **..random captions help reduce SS is an overstatement.** While we are unsure which section the reviewer is referring to, we will address this as best as we can. In all our LAION expts, we never used "truly random" captions. We always used BLIP captions which describe the contents of the image reasonably well. And training with BLIP caps (along with original caps) consistently results in lower top 95% SS.  Our statements were motivated by this consistent drop in SS.
>
> ### [Use Roberta instead of CLIP]
> We adopted CLIP for text similarity computation in Fig. 2 due to its use in SD conditioning. As per the reviewer's suggestion, we recreated Fig. 2 (of SD 1.4), using RoBERTa and we present it in **response pdf Fig 24**. The change minimally impacts results and still shows that SD v1.4 generates memorized images when both images and captions are duplicated. The Spearman Correlation b/w RoBERTa and our previous metric is **0.899**,  indicating consistent rankings. We will include the new figures for SD v1.4 and v2.1 using RoBERTA to the appendix.
>
> ### [possibly because FID is lowest when the dataset is perfectly memorized - L147-148]
> Our experiments follow Somepalli et al.'s framework, sampling inference captions from training data. We ensured no exact image or caption duplicates in our training data. In this context, if the model perfectly memorizes image-caption pairs and associates them during inference, FID -> 0 as inference images -> training size with non-repeating captions. We intend to revise this phrasing to clarify the connection.
>
> ###  [Misc Qs]
> **section 3 and Fig. 1 proper attribution** We are happy to amend section 3 and Fig. 1 that this SSCD-based approach for finding duplicates was first introduced in Somepalli et al.
>
> **Q. reg. metadata:**  Originally, we observed higher similarity within a subset of our selected parquet file compared to other slices, indicating potential metadata structure. Specifically, we used the 2nd parquet file and computed the SSCD similarity b/w the first 800K samples to the next 800K samples and found $\sim 805K$ edges above our selected threshold. When comparing the first 800K samples from the 2nd parquet file, to the first 800K samples from the 7th parquet file  we found much fewer edges $\sim 734K$. However, prompted by the reviewer, we tested this across multiple slices and observed the metadata may indeed not be structured. We will update the next draft to reflect the same. Importantly, this observation doesn't impact the analysis in Section 5.1.
>
> **Results in 5.2:** They are blip-caption conditioned models for Imagenette and original-caption conditioned for LAION-10k experiments.
>
> **Clarifications on Fig.4:** In this experiment, even though the training images are the same, the captions used in each of the experiments vary. Since we draw the captions from the training data (to study the worst-case memorization behavior), essentially the captions used in generation vary between the different caption settings.
>
> Overall, we appreciate the reviewer's insights, and these additional experiments have added depth to our analysis. We hope we addressed your questions sufficiently and we would be grateful if you would consider raising your score. Do you have any additional questions we can address?

---

> > ### Comment · Reviewer_rsTQ · 2023-08-14
> > **Response to Author comments**
> >
> > **On FID scores**: The author's response about generation quality and FID scores being uncorrelated precisely underscores my concerns about this work. At one end, you are trying to suggest that FID scores can be all over the place, and the models can still be good at generation. But at the other end, the entirety of this work is driven by comparisons of FID and SS. These results make me further worried about the paper and where the field is headed.
> >
> > **On DS scores**: I do not have a problem with SSCD, I think that is a justified metric. My concern is regarding DS. I do not think it uses SSCD in the right way, and the metric is way too uninformative. This is exemplified at multiple places in the paper, especially in the clarifications regarding Figure 18 and how duplication changes the DS. This whole discussion feels very messy and ad hoc, again alluding to the idea of easy and complex images. In this regard, can you talk more about the question in Strengths (4). With regard to the immediate question, why does duplication not influence the DS score significantly if the test sampling strategy is same? More concretely, from what I understand, duplication may not "shift" the histogram to the right because the complexity of samples stays the same, but it should still "shift" the histogram upwards, with SSCD of each sample increasing (especially for the ones that were already being memorized)---thereby also increasing DS? Can you also describe the whole setup here in detail again, I might be off from the point.
> >
> > Finally, I would request the senior authors of this paper to not mock follow-up comments by other reviewers over social media. First, this has already broken anonymity for me (my review will still stay unbiased), second I do not find it to be a respectful practice.

---

> > > ### Author Response · Authors · 2023-08-14
> > > **Clarification**
> > >
> > > Can the reviewer please confirm what they mean by "DS score"? Just to make sure we understand your comment correctly.

---

> > > > ### Comment · Reviewer_rsTQ · 2023-08-14
> > > >
> > > > DS score: dataset similarity score, as defined in L99.

---

> > > ### Author Response · Authors · 2023-08-14
> > > **Additional Clarifications**
> > >
> > > We thank the reviewer for engaging. We first want to say that we take the integrity of the review process **very** seriously.  No author of this paper has mocked any follow-up comment from this paper’s reviewers on social media.  In fact, only one other reviewer posted a follow-up comment, and that comment was normal and nice: “Thanks for your answers, I will keep my score.”  We agree with you that anonymity is important during the review period as is being supportive of reviewers, many of whom are early graduate students learning to navigate this process.
> > >
> > >
> > > **Reg. FID scores:** We do not claim that there is no correlation between generation quality and FID scores.  Note that FID scores are affected by a number of features, including diversity and image quality, which factor into the fit of the data distribution. In our previous response, we noted that the FID scores went up in one case despite maintaining comparable image quality because the model did not learn certain classes well and hence the fit of the data distribution was degraded. We further want to clarify that FID scores are not a focal point of our work.  We instead focus on memorization and only use FID scores to make sure performance does not degrade too dramatically.
> > >
> > >
> > > **Reg. DS scores:**  Regarding the trends in Figure 18, even with no training data duplication (ddf=0), the model memorizes some images (typically simple images), and as we start increasing the duplication factor, the model sees those samples, which it would otherwise memorize, less often and its memorization slowly shifts over to the duplicated samples.  Thus, it makes sense that memorization might actually decrease initially as the duplication factor goes up, namely as memorization decreases amongst samples that would have been memorized with ddf=0.  As the duplication factor increases, eventually duplicated samples will dominate, and the model will memorize them.
> > >
> > > Note that even in the case of duplicated captions (red curve in Figure 18), it is also the case that the model stops memorizing samples that would have been memorized without duplication as ddf increases, but we do not see this effect in the curve because this effect is dominated by the rapid memorization of duplicates (memorization is easier with when both image and caption are duplicated).  We will update the final draft to discuss this trend, and we appreciate your feedback.
> > >
> > >
> > > For completeness' sake, the setup is as follows: At ddf=0, the model sees all examples the same number of times during training. At ddf=2, a subset of images S (5%) is twice more likely to be sampled than the rest of the images. At higher ddf=n, this subset is now n-times more likely to be sampled. This setup controls how over-represented this subset S is across training iterations.
> > >
> > > Thank you again for your feedback. We made a significant effort to address your feedback and would appreciate it if you would consider raising your score in light of our response. Do you have any additional questions we can address?

---

### Official Review · Reviewer_qP9c · 2023-07-21

**Soundness:** 3 good
**Presentation:** 3 good
**Contribution:** 3 good
**Rating:** 6
**Confidence:** 3

**Summary:**

This paper systematically investigates the memorizing phenomenon of stable diffusion. As they empirically observed, most of the memorizing phenomenon originates from the duplication of images and captions in the training set. Owing to this, the authors propose to deduplicate the captions during the training stage to obviate the memorization of the diffusion model.

**Strengths:**

The experiments in this paper are quite complete. In my opinion, most of the components that may be related to the mesmerization phenomenon are considered.

**Weaknesses:**

To measure the memorization magnitude, the authors use a similarity score and set the threshold as 0.5, i.e., the score is larger than 0.5 means there is a memorization for a certain generated image. My concern is whether this score and threshold are valid criteria.

For the stable diffusion trained on a large-scale dataset without duplication i.e., stable diffusion v2.1, the memorization phenomenon seems almost mitigated. Thus, my question is whether increasing the unduplicated data can fix the memorization phenomenon.


**Questions:**

In Section 5.2, the authors explore the effect of the data duplication rate. They vary the rate via increase the sampled probability of duplicated data. My question is by doing so, the underlying ground-truth distribution is changed. Since the model is trained to approximate the ground-truth one. Does it reasonable for the model to generate more data similar to the duplication cluster?

What sampler do you use to generate data, DDIM or DDPM? Does the noise in DDPM helps mitigate the memorization phenomenon?

---

> ### Author Rebuttal · Authors · 2023-08-08
>
> We thank the reviewer for recognizing the merit of our work and acknowledging the thoroughness of the experiments.
>
> ### [Is SSCD score $> 0.5$ a reasonable proxy for memorization?]
>
> Somepalli et al [1] found if the similarity of SSCD embeddings of 2 images is greater than 0.5, they are most likely to be extremely similar to each other. On the fine-tuned models on Imagenette and LAION-10k, we also found that 0.5 is a reasonable cutoff while 0.7 is stricter threshold to detect potential copies. Please also refer to **Fig. 22 in the response pdf** to get a sense of how much the similarity score changes based on how similar or different the images are.
>
> ### [Can increasing the unduplicated data remove memorization?]
>
> While most of the instances in which the model memorized due to training data *exact-duplication* disappeared in SD 2.1 compared to SD 1.4, some instances like "Canvas Wall Art Print" studied in Somepalli et al [1] still exist. From our experiments, we found that even in the absence of training data duplication, some memorization exists. We observe that if an image has a unique caption (For e.g. "Pubg" in Fig 1 or "Bloodborne" from Somepalli et al. [1]), the model is a bit more likely to memorize. Also if images are too simple (as we examined in section 6, figure 7), the model can still memorize. These are a few factors we have examined and it is highly possible that there might be other underlying factors that we hope the future works will uncover.
>
>
>
> ### [Are similarity scores higher due to changes in underlying distribution?]
>
> We apologize that we missed mentioning an important detail in the paper. For a given dataset (say LAION-10k), we freeze the inference captions for all the experiments, and all the metrics computed are on generations based on these captions.  While it is true we are changing the training data for each of these models, the frozen captions ensure the computed similarity score metric is comparable across the experiments. The computed score is also a lower bound of memorization in these models. Additionally, this setting is closest to reality since we don't sample the same caption multiple times during inference even if the training data contains duplicates.
>
> ### [Impact of sampler]
> We used the DDPM sampler for training our models. We will add this detail to the experimental section. The effect of the sampler on memorization is indeed an interesting question. We leave this study for future work and we will amend our conclusion to include this question.
>
>
> We thank the reviewer for a thoughtful review. We hope we addressed your concerns/questions sufficiently and we would appreciate it if you would consider raising your score in light of our response. Please let us know if you have additional questions we can address.
>
> --------
> References:
> [1] - Somepalli, Gowthami, et al. "Diffusion art or digital forgery? investigating data replication in diffusion models." _Proceedings of the IEEE/CVF Conference on Computer Vision and Pattern Recognition_. 2023.

---

> > ### Comment · Reviewer_qP9c · 2023-08-14
> > **Reponse**
> >
> > Thanks for your answers, I will keep my score.

---

### Official Review · Reviewer_ciAp · 2023-07-25

**Soundness:** 2 fair
**Presentation:** 3 good
**Contribution:** 3 good
**Rating:** 6
**Confidence:** 3

**Summary:**

The authors conduct a large-scale empirical study about the memorization problem in diffusion models. Their experimental results indicate that text conditioning plays an important role in this problem. Based on these observations, they propose to randomize the text prompts during training to mitigate memorization and copying.

**Strengths:**

1. The memorization problem is valuable for the AI safety community.

2. The authors design a large number of experiments to study this problem.

3. The authors observe that text conditioning plays an important role in this problem, which is a new discovery compared with the existing works.


**Weaknesses:**

1. Some experiment designs cannot verify the arguments. And some results are not consistent with the arguments of the authors. Please refer to Question1-3.

2. The proposed solutions are not novel. Please refer to Question4.

3. Some discussion and references about mode collapse are missing. Please refer to Question5.

4. Some figures in the paper are messy and confusing. Please refer to Question6.



**Questions:**

- 1. In line177, the conclusion ‘’the model is more likely to memorize images when the captions are more diverse” conflicts with the observations in Section 7. The authors propose several data augmentation methods to make the text prompts as diverse as possible, which successfully mitigates the memorization problem. This indicates that when the captions are more diverse, the model is less likely to memorize images. Please clarify this conflict.

- 2. In line185-186, why do you draw a conclusion that ‘’the model is more inclined to remember instances when the captions associated with them are highly specific, or even unique, keys” according to the experiment phenomenon that the similarity score of random captioning increases after finetuning the text encoder? Wouldn't random captions be even less SPECIFIC keys?

- 3. Please report the FID values in Table1 to verify whether the proposed solutions will hurt the generation quality.

- 4. The mitigation strategies in Section7 are not novel. Even though some of them have positive effects, they are just some data augmentation tricks that should have been used.

- 5. Actually, I find the memorization problem discussed in this paper is related to a common problem in generative models – mode collapse. Especially in the failure cases mentioned in line149-157, the phenomenon is very similar to mode collapse – the generative models perform terribly when they are only able to cover a small portion of the data distribution. As for the memorization problem in Stable Diffusion, I think it is a special case of mode collapse that the text conditioning model collapses because there are not diverse enough text prompts in the training set, which is consistent with the analysis and the proposed mitigation strategies in this paper. Please discuss the difference between memory problems and mode collapse, if you disagree with me, and vice versa, please mention the connection in the related work section.

- 6. Some figures are confusing. For example:

    - a) There is no text prompt in Figure1, so it is hard to understand the error modes. Take the 4-th image as an example: if the text is ''The front page of PUBG”, the output image should be the 4-th image in the lower row and thus I don’t think this is a problem.

    - b) Please use the vector figures in the right two subfigures in Figure2. They look very blurry.

    - c) There is no caption about the left subfigure in Figure2. I don’t know why you put it here.

    - d) There is no legend for the two dash lines in Figure4. And I suggest drawing the FID scores of finetuned models above the chart bars rather than writing them in the caption.

**Limitations:**

The authors do not discuss the limitations.

---

> ### Author Rebuttal · Authors · 2023-08-09
>
> We thank the reviewer for recognizing the significance of the memorization problem in the AI safety domain. We sincerely appreciate your acknowledgment of our extensive experimental design.
>
> ### [Clarifications on caption diversity]
>
> We apologize that our language at L177 is slightly confusing. When we said "... model is more likely to memorize images when the captions are more diverse", we meant that as the caption gets more specific, the model is more likely to remember that sample. For example, let's say there is an image of a sunset. A less diverse caption would be "A beautiful sunset" while a more diverse/specific caption would be "A sunset in Hawaii shot by Ansel Adams". Another way to put it is, as the perplexity of caption (wrt model trained on general English text as well models trained with the captions of training dataset) is high, the model is more likely to remember the corresponding image.
>
> Even though with less specific (low perplexity) captions, we observe low memorization, at the same time, we sacrifice the quality of generations (with higher FID scores). In section 7, we suggest using multiple captions instead of one caption for each image to break the one-to-one image-caption correspondence. This ensures that the quality of generations stays the same while reducing memorization.
>
>  We will replace the word "diverse" with "specific" or "high perplexity" in L 177-178, clarify the language, and add a small paragraph at the beginning of section 5 defining these keywords.
>
> ### [L185-186, Clarifications on caption specificity]
>
> Following the previous example, for the same sunset image, if the caption is random, such as "guatemala muenvica gregation", this caption is much more specific than the "A sunset in Hawaii shot by Ansel Adams" caption and the model is even more likely to remember (as shown in Fig. 4 (Right)). We see empirically that as the captions get more specific (high perplexity) if the text encoder is aligned with the captions, we see the image quality improves or stays the same while memorization goes up. (correlation coeff = -0.63 between FID and SS for Fig. 4, right, when text encoder is trained).
>
> We will update the draft with an example so that the readers will get a sense of caption specificity in these experiments.
>
> ### [FID scores for the mitigation strategies]
> We have provided the FID scores for mitigation strategies in Table 3 in Appendix. We see that the models trained with the Multiple Captions mitigation strategy **always** had lower FID scores along with lower memorization. Please note that many mitigation strategies resulted in lower FID scores.
>
> ### [Mitigation strategies are not novel]
> We respectfully disagree with this comment. While some of these techniques may have been previously used as augmentations elsewhere, their application in training diffusion models to mitigate memorization has been unexplored until now. Further, the mitigations we select are based on our empirical findings from the initial part of the paper. We believe these findings contribute to the advancement of our understanding of memorization in diffusion models and are in turn validated by practical innovations for those training diffusion models on sensitive data.
>
> ### [Memorization vs Mode Collapse]
> This is a very intriguing observation. From our current perspective, mode collapse and memorization are distinct phenomena in the context of deep learning models, although both can be considered failure modes. In the GAN literature, mode collapse is defined as the generator failing to capture the full data distribution, producing a limited range of similar samples and neglecting distribution diversity. This is a result of joint training of generator, decoder, and encoder.
>
> Conversely, memorization aligns closely with overfitting, as indicated in [1,2]. From this viewpoint, the model cannot simultaneously fit the target distribution too precisely (overfitting/memorization) and collapse to only a portion of it (mode collapse). An interesting conjecture is that mode collapse might resemble strong overfitting to specific examples. That said, your proposition regarding mode collapse as a mechanism for memorization is captivating. However, a direct collapse akin to GAN encoder mode collapse isn't feasible in the text-to-image diffusion models we analyze, where the text encoder isn't trained.
>
> Yet, determining a clear, falsifiable hypothesis from this connection poses challenges. Could you suggest a specific mode collapse definition isolating this effect? We're open to trying your suggestion. Regardless, we'd be pleased to incorporate this potential link into our related work.
>
> ### [Misc improvements]
>
> **Reg. PUBG case.** While it is a legal question whether a model can or cannot generate an image, we follow the same definition of memorization from [3] which follows the most stringent definition of memorization where the model cannot generate anything that a human will perceive as a copy. In Fig. 1, even though the PUBG images are not exact copies, they are close enough to be perceived as copies by humans, and while we cannot speak with legal certainty, using the generated image as a game logo, for example, would imply a sufficient violation of the PUBG logo trademark.
>
> **Reg. figures.** We thank the reviewer for taking the time to suggest improvements to our figures to make the paper more accessible. We will make these changes for the camera-ready version.
>
> Thank you for an insightful review with important questions and meaningful suggestions. We hope we addressed your questions sufficiently and we would appreciate it if you would consider raising your score in light of our response. Please let us know if you have additional clarifications.
>
> ---
>
> [1] Carlini et al. "The secret sharer" USENIX Security 19
>
> [2] Feldman et al. "Discovering the long tail via influence estimation." NeurIPS'20.
>
> [3] Somepalli, et al. "Diffusion art or digital forgery?" CVPR'23

---

> > ### Comment · Reviewer_ciAp · 2023-08-15
> > **Thanks for your rebuttal**
> >
> > I appreciate the response from the authors. I think my concerns have been addressed. I improve the score. Please remember to have the clarifications on caption diversity and specificity in your next version. BTW, I still worry that the statement "... model is more likely to memorize images when the captions are more diverse" will confuse the readers. Please consider revising this paragraph.

---

> > > ### Author Response · Authors · 2023-08-15
> > > **Thank you**
> > >
> > > Thank you again for your feedback.  We will most certainly include that clarification in our camera-ready version.

---

### Author Rebuttal · Authors · 2023-08-09

We would like to thank **all six reviewers** for their highly constructive feedback! We very much appreciate their assessment that we are "studying an important problem" (ciAp, 4RBY, R9Bm),  with "thorough experimentation" (ciAp, qP9c, rsTQ, 4RBY, MEzn, R9Bm),  "first to introduce text conditioning.." (ciAp, 4RBY, MEzn), with "results are valuable to community" (rsTQ),  "thought-provoking" (rsTQ), and "a great starting point... memorization" (rsTQ,4RBY),  and we raised "interesting image complexity argument" (rsTQ, 4RBY),  and finally it is "well written" (4RBY, MEzn, R9Bm).

The reviewers raised many excellent and thought-provoking questions and we conducted many experiments to address these questions. This interaction definitely added more depth to the paper and helped us understand this phenomenon better.

### [A brief summary of questions and additional results]
1. **Choice of 95th percentile for similarity metric**: To clarify the rationale behind selecting the 95th percentile over 99 or 50, we added an ablation in Fig 17.
2. **SSCD similarity score for memorization** We added Fig 22 which shows how generation & top matches look like at different SSCD scores.
3. **Range of similarity scores and what it means?** We conducted additional experiments (with 20\% of data duplicated) and presented those results in Fig 19. We see that when the amount of duplicated training data increases, sim score metric's range increases.
4. **Partial duplication further analysis**  We took a deep dive into the question of why partial duplication has lower SS than the model with no duplication. We found that a slight domain shift between training and testing led to this, and recomputed these results and presented them in Fig 18. The original findings still hold.
4. **Number of copies** - We computed the actual numbers of images with SS > 0.05 and presented the results in a table as part of our response to **reviewer 4RBY**. This additional analysis also shows that there is a significant reduction of such images with the MC mitigation strategy.
5. **Diversity metrics** For completeness we will add precision, and recall metrics to the paper. One such plot is shown in Fig. 20 (right).
6. **How good are ImageNette models** We presented a few generations of text-encoder-frozen IN models trained with different styles of captions in Fig 21. We will add an expanded version of this figure to the appendix.
7. **Qual results for same images for both test and train time mitigation** Because the training data was fixed before we found the memorized prompts in SD 2.1, we chose different examples to show the effect of train/test time mitigation strategies. For the response, we trained a few new models adding these 5 images to the train set. We recreated the train-time qualitative analysis figure as shown in Fig 23.
8. **Roberta instead of CLIP for image cap analysis** We recreated the main paper Fig 2 using Roberta and presented it in Fig 24. The findings stay the same.
9.  Final draft will incorporate these enhancements: limitations paragraph in conclusion clarifying experiment scope and metric. Introduction section will reemphasize paper's memorization scope. Related work will cover mem. vs mode collapse. Address typos, readability, define with examples, improve figures, add missing citations.

### [Paper's definition of memorization]
In this paper, we follow the same definition of memorization as Somepalli et al where a reasonable resemblance (by a human) to a specific entity is considered a copy. Even when the prompt explicitly references a brand logo and the model duly generates that logo, it is unequivocally categorized as a memorization instance.  The legality of whether a generation is fair use or not is beyond the scope of this work.

### [Is SSCD good for understanding memorization?]
We have found that similarity scores above 0.5 empirically are strongly indicative of memorization. We added a figure to **response pdf, Fig 22** that shows representative image pairs with different levels of sim score. One can see that above 0.5, image pairs appear to be almost copies of one another.  Note that all these examples are from the LAION-10k model trained with no duplication.

### [The choice of percentile in Similarity Score metric]
Occasional memorization, even in a small percentage of generations, can pose problems—precisely the pattern we've observed. To capture this behavior effectively, we measure the sim score of the top 95% of samples. Using the average or median, in contrast, could fail to detect significant memorization in a few instances. Our choice of the 95% percentile is based on observations by Somepalli et al., who identified around 2% of memorization. Adjusting for fine-tuning on a smaller dataset, we found 95% to be a suitable benchmark.

In response to your query, we've performed an ablation analysis on LAION-10k and full duplication across different ddf settings. Results, as displayed in **Fig 17 of our response pdf**, follow same trends across 95, 97, and 99 percentiles. Median scores remain relatively constant, as expected, due to memorization affecting only a small percentage of images, as previously noted by Somepalli et al. Similar trends appear in partial duplication experiments, as evident in Fig 25 (left). We acknowledge that we needed to justify this design choice better, we will add a paragraph on the motivation behind this design choice in the experiments section.

---
In conclusion, we wish to underscore the current relevance and substantial significance of this research. Since the SD lawsuit, addressing memorization became pivotal for companies. Many recent papers offer short-term solutions, but our study delves deeper. We comprehensively examined underlying factors and proposed mitigation strategies based on what we learned from our analysis. Since the paper came out, an industry-leading model provider has reached out for guidance on how to implement our methods in their pipeline.

---

### Decision · Program_Chairs · 2023-09-21

**Decision:**

Accept (poster)

**Comment:**

This work presents an in-depth empirical study of the memorization problem of text-to-image diffusion models. Reviewers are overall positive about this work, and think this work is timely, novel, important, and the experiments are comprehensive and conclusive. One major concern is the metric (dataset similarity scores) used to measure memorization, which requires further motivation and justification. AC agrees that this work is a welcome practice towards addressing the memorization and copying challenges of modern generative AI, and thus recommends acceptance.